# Phylogenomic, Morphological, and Phylogenetic Evidence Reveals Five New Species and Two New Host Records of *Nectriaceae* (*Hypocreales*) from China

**DOI:** 10.3390/biology14070871

**Published:** 2025-07-17

**Authors:** Qi Fan, Pingping Su, Jiachen Xiao, Fangwei Lou, Xiaoyuan Huang, Zhuliang Yang, Baozheng Chen, Peihong Shen, Yuanbing Wang

**Affiliations:** 1College of Life Science and Technology, Guangxi University, Nanning 530004, China; fanqi_666@163.com (Q.F.);; 2State Key Laboratory of Phytochemistry and Natural Medicines, Kunming Institute of Botany, University of Chinese Academy of Sciences, 132 Lanhei Road, Kunming 650201, China; 3CAS Key Laboratory for Plant Diversity and Biogeography of East Asia, Kunming Institute of Botany, Chinese Academy of Sciences, Kunming 650201, China; 4Yunnan Key Laboratory for Fungal Diversity and Green Development, Kunming Institute of Botany, Chinese Academy of Sciences, Kunming 650201, China

**Keywords:** *Fusarium*, *Neocosmospora*, Endophytic fungi, phylogenomics, taxonomy

## Abstract

*Fusarium* and *Neocosmospora* are widely distributed in natural environments and include species that are both beneficial and pathogenic. However, their overlapping morphological and ecological characteristics have posed a challenge to the delimitation of species. In this study, 22 fungal strains, isolated from diverse hosts including plants, insects, and entomopathogenic fungi, were analyzed using an integrative approach that combined morphological and molecular data. As a result, five new species were described, and two new host record species were reported. Phylogenomic evidence further confirmed the distinct taxonomic boundaries between *Fusarium* and *Neocosmospora*. These findings contribute to our understanding of the biodiversity and evolution of fusarioid fungi.

## 1. Introduction

The *Nectriaceae* (*Ascomycota*, *Hypocreales*) is a globally distributed and taxonomically diverse fungal lineage characterized by complex phylogenetic relationships. Members of this family exhibit diverse ecological lifestyles, including saprotrophs, endophytes, and plant, animal, and fungal pathogens [1,2,3,4,5,6,7,8]. In this family, species exhibiting fusarium-like asexual morphs are commonly referred to as fusarioid fungi (https://www.fusarium.org/, accessed on 28 May 2025) [4]. Among them, *Fusarium* and *Neocosmospora* are the largest and most intensively studied. These genera play essential ecological roles in natural ecosystems and are also associated with numerous economically important plant diseases [6,9]. Moreover, several species within these genera are recognized as opportunistic human pathogens [10,11,12]. Therefore, the fusarioid fungi have attracted sustained attention in fungal taxonomy, plant pathology, and medical mycology.

The *Fusarium* was established by Link in 1809 to accommodate fungi characterized by falcate macroconidia [13]. Since its original description, the genus has undergone numerous significant revisions to its taxonomic framework and nomenclatural system, often accompanied by ongoing controversy and conflicting taxonomic approaches [8,14,15,16,17,18,19,20,21,22]. Crous et al. [4] conducted a comprehensive taxonomic revision of fusarioid taxa within the *Nectriaceae*, integrating multigene phylogenetic analyses with morphological characteristics. Their study robustly supported the delimitation of 20 distinct fusarioid genera, based on the Wollenweber classification system, and explicitly supported a narrow generic concept of *Fusarium* (=*Gibberella* = “F3 clade”). Building upon this phylogenetic framework, Wang et al. [23] and Han et al. [6] further updated the taxonomy of *Fusarium* sensu stricto. At present, *Fusarium* s. s. comprises 18 recognized species complexes and one undefined complex, represented by *F. nurragi* [4,5]. Nevertheless, some researchers hold different views regarding this taxonomic treatment. They advocate maintaining a broad definition of the *Fusarium* (=“F1 clade”) to avoid introducing additional genera, thereby retaining these agriculturally and medically important taxa within *Fusarium* [17,18]. The divergence in taxonomic perspectives and the ensuing debates reflect the complexity of *Fusarium* taxonomy and have, to some extent, stimulated deeper phylogenetic and functional investigations of the genus.

The genus *Neocosmospora* was established by Smith in 1899, with *N. vasinfecta* E.F. Sm designated as the type species [24]. The taxonomic placement of this group has also been subject to debate, particularly concerning its delimitation from the genus *Fusarium*. Sandoval-Denis et al. [8] re-evaluated *Neocosmospora* based on morphological characteristics and phylogenetic analyses and proposed that the *F. solani* species complex (FSSC) should be transferred to this genus. In contrast, O’Donnell [18] and Geiser [19] advocated for retaining the FSSC within the *Fusarium*, emphasizing the phylogenetic coherence of the broader *Fusarium* lineage. However, Crous et al., through an extensive investigation of *Fusarium* and related genera, supported the taxonomic transfer of the FSSC to *Neocosmospora* [4]. Currently, the FSSC is widely recognized by the scientific community as a member of the genus *Neocosmospora* [3,4,5,25,26,27,28,29,30].

Accurate species identification is the primary prerequisite for the diagnosis and effective management of diseases caused by fusarioid fungi [6,31,32]. Given the complexity of species delimitation and identification within fusarioid fungi, researchers have increasingly adopted a polyphasic taxonomic approach, integrating morphological, phylogenetic, and ecological data to conduct comprehensive taxonomic studies [4,5,6,23,33,34]. In particular, extensive studies have been conducted on major species complexes such as the *F. sambucinum*, *F. fujikuroi*, and *F. oxysporum* species complexes [35,36,37,38]. These studies highlight the necessity of using lineage-specific markers to achieve accurate species-level resolution [5,6,23,32]. With the rapid advancement of high-throughput sequencing technologies, phylogenomics has demonstrated significant advantages in elucidating evolutionary processes and delineating species boundaries. This approach has been increasingly applied to integrative studies of fungal phylogeny and pathogenicity [39,40,41,42]. In the case of fusarioid fungi, phylogenomic analyses have also been employed to assess intergeneric boundaries, providing strong support for the refinement of its taxonomic framework [6,19,43,44].

In this study, 22 fungal isolates were obtained from Zhejiang and Yunnan provinces in China. To clarify their taxonomic positions and phylogenetic relationships, we employed an integrative approach combining multilocus phylogenetic analyses with morphological characterization. Additionally, to investigate the phylogenetic boundaries between the genera *Fusarium* and *Neocosmospora*, whole-genome sequencing and analyses were conducted on seven representative isolates. These efforts aim to provide novel insights and molecular evidence that contribute to resolving the evolutionary relationships and species delimitation within fusarioid fungi.

## 2. Materials and Methods

### 2.1. Sample Collection, Strain Isolation, and Preservation

Samples were collected from Yunnan and Zhejiang provinces, China, including symptomatic stems of *Musa* sp., fungus-infected adult *Lepidoptera*, asymptomatic cordycipitoid fungi, and leaves of *Dracaena* species. Pure cultures were obtained using single-spore isolation from the *Lepidoptera* samples, while tissue isolation was applied to the remaining three sample types [5,45,46].

Tissue isolation techniques: First, the samples were rinsed with sterile water to remove surface impurities. Then, they were immersed in 75% ethanol (Sangon Biotech Co., Ltd., Shanghai, China) for 10 s for surface sterilization. Subsequently, the samples were rinsed with sterile water for 30 s and disinfected by soaking in 30% hydrogen peroxide (Sangon Biotech Co., Ltd., Shanghai, China) for 1 min. Afterward, they were rinsed three times with sterile water (each rinse lasting 30 s) and blotted dry with sterile filter paper. Finally, the samples were cut into small segments approximately 5 × 5 mm^2^ in size for further use. The tissue fragments were then transferred to potato dextrose agar (PDA) medium (200 g/L potato, 20 g/L dextrose, 18 g/L agar) supplemented with 100 mg/L penicillin (Sangon Biotech Co., Ltd., Shanghai, China) and 100 mg/L streptomycin (Sangon Biotech Co., Ltd., Shanghai, China). Plates were incubated at 25 °C for 3 days. Hyphal tips were excised from the inoculation site and transferred to fresh PDA medium for initial purification. Subsequently, hyphal tips from the margins of the newly developed colonies were aseptically transferred to fresh plates and subcultured for 2–3 successive rounds to eliminate potential contaminants. The final cultures were examined microscopically to verify the purity of the isolates. Single-spore isolation method: To obtain monosporic cultures, spores were inoculated onto PDA medium using an inoculation needle and incubated at 25 °C. After approximately two days, actively growing hyphae from the colony margins were transferred to fresh PDA for purification. Once the colonies fully covered the PDA and spore production was confirmed under an Olympus BX53 microscope (Olympus Corporation, Tokyo, Japan), spores were harvested by rinsing the colony surface with sterile water. A spore suspension was then prepared at a concentration of 1 × 10^3^ spores/mL. Subsequently, spore suspensions (20 μL per plate) were inoculated onto PDA to obtain monospore cultures. The purified cultures were subsequently transferred to PDA slants and maintained at 4 °C for storage. Specimens were deposited in the Cryptogamic Herbarium of the Kunming Institute of Botany, Chinese Academy of Sciences (KUN-HKAS). Cultures were deposited in the Kunming Institute of Botany Culture Collection (KUNCC), Chinese Academy of Sciences.

### 2.2. DNA Extraction and PCR Amplification

Genomic DNA was extracted from fresh mycelia cultured for three weeks using the Ezup Column Fungi Genomic DNA Extraction Kit (Sangon Biotech, Shanghai, China), following the manufacturer’s protocol. PCR reactions were performed using a LongGene T20 multi-block thermal cycler (Hangzhou LongGene Scientific Instruments Co., Ltd., Hangzhou, China). Each 25 µL reaction mixture contained 12.5 µL of 2× Taq PCR Master Mix (Tiangen Biotech Co., Ltd., Beijing, China), 9.5 µL of RNase-free water (Sangon Biotech Co., Ltd., Shanghai, China), 1 µL of each forward and reverse primer (10 µmol/L), and 1 µL of DNA template (500 ng/µL).

Primers were selected following Crous et al. and Han et al. [4,6]. For preliminary Maximum Likelihood (ML) phylogenetic analyses, translation elongation factor 1-alpha (*tef1*) and RNA polymerase II second largest subunit (*rpb2*) were amplified and sequenced. Subsequently, primers specific to each taxonomic group were selected for further phylogenetic analysis. For the isolates assigned to *Fusarium*, internal transcribed spacer (ITS), *rpb2*, RNA polymerase II largest subunit (*rpb1*), *tef1*, and beta tubulin (*tub2*) were used for the *F. heterosporum* species complex (FHSC); calmodulin (*CaM*), *rpb2*, *rpb1*, and *tef1* were used for the *F. incarnatum-equiseti* species complex (FIESC), while *CaM*, *rpb2*, *tef1*, and *tub2* were used for the *F. lateritium* species complex (FLSC). For isolates assigned to the *Neocosmospora*, the loci ITS, *tef1*, *rpb1*, *rpb2*, ATP citrate lyase (*acl1*), and tub were selected for amplification and phylogenetic analysis. Detailed information on the primer pairs and PCR cycling conditions is provided in Appendix A. Standard DNA markers (Sangon Bio Co., Ltd., Shanghai, China) of known size were used to estimate fragment length. Sanger sequencing was conducted by Sangon Biotechnology Co., Ltd. (Kunming, China) and Tsingke Biotechnology Co., Ltd. (Kunming, China).

### 2.3. Whole-Genome Sequencing, Assembly, and Gene Annotation

Whole-genome sequencing was conducted for five ex-type strains of new species (*F. dracaenophilum*, *F. puerense*, *F. wenshanense*, *F. fungicola*, and *F. alboflava*), as well as for two strains of known species (*F. qiannanense* and *N. solani*). All strains were cultured in Potato Dextrose Broth (PDB; potato 200 g/L, dextrose 20 g/L) for five days. Fresh mycelia were harvested, immediately frozen in liquid nitrogen, and then stored at −80 °C for subsequent genomic DNA extraction. An amount of 1 µg of DNA per sample was used as input for library construction. Sequencing libraries were prepared using the VAHTS Universal DNA Library Prep Kit for MGI (Vazyme, Nanjing, China) following the manufacturer’s protocol, with index codes added to assign reads to individual samples. Library concentration and fragment size were assessed using a Qubit 3.0 Fluorometer (Life Technologies, Carlsbad, CA, USA) and a Bioanalyzer 2100 system (Agilent Technologies, Santa Clara, CA, USA), respectively. A paired-end library with an average insert size of 350 bp was generated using the GenElute Plant Genomic DNA Miniprep Kit (Sigma-Aldrich, St. Louis, MO, USA), according to the manufacturer’s instructions. Sequencing was performed on a DNBSEQ-T7 (BGI, Beijing, China) platform by Frasergen Bioinformatics Co., Ltd. (Wuhan, China).

Quality assessment of the raw DNBSEQ short-read data was performed with FastQC v0.12.1 [47], followed by quality filtering using Fastp v0.23.4 [48], which automatically detects and removes adapter sequences and low-quality reads. Genome assembly was conducted using SPAdes v3.12.0 [49], and its quality was assessed using QUAST v. 5.0.2 [50]. Assembly completeness was estimated using BUSCO v5.5.0 with the lineage-specific profile library hypocreales_odb10 [51,52]. Gene predictions and annotation of all assemblies, including the downloaded outgroups, were performed with the funannotate pipeline v1.8.4 (https://funannotate.readthedocs.io/, accessed on 1 April 2025).

### 2.4. Phylogenetic Analyses

Multigene phylogenetic analysis refers to the reconstruction of phylogenetic relationships among species or populations based on multiple (typically 2 to 20) independently evolving genes or gene fragments, using concatenation or consensus-based approaches [3,4,6,23,39]. In contrast, phylogenomic analysis involves the use of high-throughput sequencing technologies to obtain large-scale datasets, usually comprising hundreds to thousands of genes or genomic regions, which are jointly analyzed to infer species relationships with greater accuracy [6,44]. Phylogenomic methods significantly improve the resolution and statistical support of phylogenetic trees, effectively addressing common limitations of multilocus analyses such as low node support and unstable topologies.

#### 2.4.1. Multigene Phylogenetic Analysis

Sequence quality was assessed using MEGA v7.0 [53], and consensus sequences were assembled with SeqMan (Lasergene v14.1, DNASTAR, Madison, WI, USA). Sequence alignments for each locus were performed using MAFFT v7 [54], followed by manual adjustments in MEGA v7.0 [53]. Ambiguously aligned regions were manually filtered, and gap characters were treated as missing. Multilocus phylogenetic analyses were conducted for three *Fusarium* species complexes and the *Neocosmospora*, following methodologies established in previous studies [6,16,35,55,56,57,58,59,60,61,62]. Specifically, for the FHSC, the analyses were based on a concatenated dataset comprising the ITS, *rpb2*, *rpb1*, *tef1*, and *tub2* loci. For the FIESC, phylogenetic trees were constructed using sequences of the *CaM*, *rpb2*, *rpb1*, and *tef1* loci. The FLSC was analyzed using a combined dataset of *CaM*, *tef1*, *rpb2*, and *tub2*. For the *Neocosmospora*, phylogenetic analyses were performed using a multilocus dataset comprising ITS, *CaM*, *rpb2*, *rpb1*, *tef1*, and *acl1*. In the *Fusarium* analysis, *Cyanonecia cyanostoma* CBS 101734 ET was uniformly designated as an outgroup, while in the *Neocosmospora* analysis, *Setofusarium setosum* CBS 635.92 ET was designated as an outgroup. ModelFinder [63] was used to select the best-fit nucleotide substitution models. Model selection for ML analyses was based on the Akaike Information Criterion (AIC), while the Bayesian Information Criterion (BIC) was applied to determine the optimal models for Bayesian Inference (BI) analyses. The composition of the multilocus datasets, the number of nucleotide positions, and the best-fit substitution models are summarized in Appendix A.

Partitioned ML and BI analyses were performed: The BI analyses were carried out using MrBayes v. 3.2 [64]. Four simultaneous Markov Chain Monte Carlo chains were run for 20 million generations, with a sampling frequency of every 100 generations. The run was automatically terminated when the standard deviation of split frequencies dropped below 0.01. A burn-in of the first 25% of the total samples was discarded, after which the 50% majority-rule consensus trees and posterior probability (PP) values were calculated. The ML analyses were conducted using IQ-TREE v. 2.1.3 [65] under partitioned models [66] with 1000 ultrafast bootstrap replicates [67]. A clade was considered well supported when its ML bootstrap value was ≥85% and its Bayesian posterior probability (PP) was ≥0.9 [68]. Phylogenetic trees were visualized with ML bootstrap proportions (ML-BS) and Bayesian posterior probability (BI-PP) using FigTree v. 1.4.4 and subsequently edited with Adobe Illustrator CS6.0. All sequences generated in this study were deposited in GenBank (Appendix A).

#### 2.4.2. Phylogenomic Analysis

The orthologous genes, both single-copy and orthogroups, in the 21 genomes were identified using a phylogeny-based orthology inference approach implemented in OrthoFinder 2.5.2 [69], with DIAMOND for sequence similarity search and local alignment and the DendroBLAST algorithm for gene tree inference. Detailed information on the genome data newly generated in this study, as well as that retrieved from the NCBI database, is provided in Appendix A. Each of the resulting single-copy orthologous gene sets was aligned using MAFFT v7 [54] with the -auto option. Subsequently, the corresponding coding sequences (CDSs) were transferred to the codon alignment according to the alignments of these protein-coding sequences using PAL2NAL v14 [70]. The poorly aligned regions within these aligned sequences were then filtered out using trimAl v1.4.rev15 [71] with the parameter “-automated1”. Finally, those alignments of the orthologous groups were concatenated and utilized to build an ML phylogenetic tree using IQ-TREE v. 2.1.3 [65] with the parameters “-m MFP; -bb 1000; -nt 10” and the best-fit model (GTR + F + I + G4). Divergence times were estimated based on the ML tree using MCMCTree v4.10.0 [72] from the PAML v.4.9h package with parameters of “burnin = 50,000; nsample = 100,000”. The calibration point between the genera *Geejayessia* and *Neocosmospora* (20.54–86.06 million years ago, Mya) represents a secondary calibration derived from Lizcano Salas et al. [44], which is itself based on the dated fungal phylogeny constructed by Lutzoni et al. [73]. In their study, 13 fossil constraints were fixed in BEAST to anchor key nodes in the Ascomycota phylogeny, including *Palaeopyrenomycites devonicus* (–400 Mya, Devonian), *Archaeomarasmius leggettii* (–90 Mya, Late Cretaceous), and *Colletotrichum* (–65.2 Mya, Upper Cretaceous). Based on this calibrated tree, Lutzoni et al. [73] applied a soft prior of 50–90 Mya at the *Fusarium*-*Neocosmospora* node in their MCMCTree analysis, and the combination of this prior with genomic substitution rate data yielded the 20.54–86.06 Mya credibility interval adopted in our analysis. The phylogenetic tree, including the divergence times, was visualized using FigTree v1.4.4 (http://tree.bio.ed.ac.uk/software/figtree/, accessed on 14 April 2025).

### 2.5. Genealogical Concordance Phylogenetic Species Recognition Analyses

The pairwise homology index (PHI, Φw) test, based on the Genealogical Concordance Phylogenetic Species Recognition (GCPSR) principle, was employed to delimit phylogenetically related but taxonomically ambiguous species. The PHI test was conducted in SplitsTree v. 6.3.27 to evaluate the extent of recombination among closely related species using a concatenated multilocus dataset [74]. To visualize these relationships, split networks were constructed using the LogDet transformation and split decomposition methods. A PHI value below 0.05 (*p* < 0.05) was interpreted as evidence of significant recombination within the dataset. Accordingly, homologous relationships between the newly described species and their phylogenetically related taxa were assessed.

### 2.6. Morphological Observations

To study the morphological characteristics of fungal isolates, both macroscopic and microscopic features were examined [4]. Colony morphology was observed using three culture media: PDA, synthetic nutrient-poor agar (SNA; 20 g/L agar, 1.0 g/L KH_2_PO_4_, 1.0 g/L KNO_3_, 0.5 g/L MgSO_4_·7H_2_O, 0.5 g/L KCl, and 0.2 g/L glucose; pH = 7.0), and oatmeal agar (OA; 30.0 g/L oatmeal flakes and 15.0 g/L agar). The isolates were initially inoculated onto PDA plates and incubated at 25 °C for seven days. Mycelial plugs (approximately 5 × 5 mm) were taken from the margins of the colonies and transferred to the three media. After incubation in the dark for seven days, colony characteristics such as pigmentation and odor were recorded [4].

For microscopic observations, mycelial plugs were transferred to carnation leaf agar (CLA) [75] and incubated at 25 °C under a 12 h near-ultraviolet light/dark cycle for 7–14 days. Sporodochia were initially examined and photographed using an Olympus SZ60 stereomicroscope (Olympus Corporation, Tokyo, Japan). Subsequently, water was used as the mounting medium, and the structures were observed using an Olympus BX53 microscope (Olympus Corporation, Tokyo, Japan) with differential interference contrast (DIC) optics. The following structures were observed: sporodochia and sporodochial conidiophores, phialides, and conidia; aerial conidiophores, phialides, and conidia; and chlamydospores [4,76].

## 3. Results

### 3.1. Molecular Phylogeny

Preliminary phylogenetic analysis based on the combined *tef1*, *rpb1*, and *rpb2* loci revealed a tree topology that broadly resembled the phylogeny proposed by Crous et al. [4]. Among the 22 representative strains, 10 were assigned to three species complexes within the *Fusarium* (i.e., FHSC, FIESC, and FLSC), whereas the remaining strains were grouped within the *Neocosmospora* (Appendix A). Subsequently, phylogenetic analyses were conducted separately for each *Fusarium* species complex and for *Neocosmospora*, using different datasets and the best-fit substitution models selected for each gene partition.

A phylogenetic tree of the FHSC was constructed based on a concatenated dataset of ITS, *tef1*, *rpb1*, *rpb2*, and *tub2* sequences from nine strains (Figure 1). The resulting tree topology was consistent with previous studies [5,6]. Similarly, the FIESC tree was generated using concatenated *CaM*, *rpb2*, and *tef1* sequence data from 68 strains (Figure 2), and similar phylogenetic results were observed [6]. For the FLSC, a phylogenetic tree was inferred by combining *CaM*, *tef1*, *rpb2*, and *tub2* sequences from 28 strains (Figure 3), and its topology was similar to that of Wang et al. [23]. Finally, phylogenetic analyses of *Neocosmospora* were performed based on combined sequences of *acl1*, *CaM*, *ITS*, *rpb1*, *rpb2*, and *tef1* from 42 strains (Figure 4), which were topologically consistent with previous studies [4,5]. Multigene phylogenetic analyses and GCPSR analyses (Figure 5), combined with morphological characteristics, revealed that the 22 isolates represent seven species, including five new and two known species. Specifically, one species was assigned to the FHSC (Figure 1), two species to the FIESC (Figure 2), one species to the FLSC (Figure 3), and three species to *Neocosmospora* (Figure 4).

Furthermore, single-locus phylogenetic trees were constructed for the three species complexes within *Fusarium* and for the *Neocosmospora*, respectively (Appendix A). Phylogenetic analyses based on single-gene loci showed that *rpb2* and *tef1* provided higher resolution for species delimitation within three *Fusarium* species complexes (FHSC, FIESC, and FLSC) and *Neocosmospora*. Specifically, in the FHSC, both *tef1* and *rpb2* achieved complete species resolution (100%, 3/3). In the FIESC, *tef1* resolved all 63 species (100%, 63/63), whereas *rpb2* resolved 50 species (86%, 50/58). In the FLSC, both loci achieved full resolution (22 of 22). In *Neocosmospora*, *tef1* resolved 29 species (94%, 29/32), while *rpb2* resolved 23 species (77%, 23/30).

### 3.2. Genomic Features

The genome sizes of the *Fusarium* and *Neocosmospora* species used in this study ranged from 34 to 54 Mb. *Geejayessia zealandicum* NRRL 22465 had the smallest assembly size (34 Mb), while *N. vasinfecta* NRRL 22166 had the largest assembly size (54 Mb). The BUSCO completeness of assemblies was in the range of 93.1% to 97.6%, with duplicated gene content ranging from 0.2% to 0.8%. The total number of predicted protein-coding genes ranged from 9479 in *G. zealandicum* NRRL 22465 to 14826 in *N. vasinfecta* NRRL 22166. Smaller genomes tended to have fewer predicted protein-coding genes (Appendix A).

### 3.3. Results of Phylogenomic Analysis

We sequenced the genomes of the seven species described in this study and included an additional 15 published genomes retrieved from the NCBI Datasets repository (https://www.ncbi.nlm.nih.gov/datasets/, accessed on 16 March 2025) for phylogenomic and comparative analysis. Using OrthoFinder v2.5.2 [69], a total of 259,957 genes from 21 genomes were clustered into 256,187 orthologous groups, with 3,770 genes remaining unclustered. Among these, 6,055 orthologous groups were shared across all 21 species, including 4,941 single-copy orthologous groups (Appendix A). An ML phylogenetic tree was constructed based on these 4,941 clusters of orthologous proteins (Figure 6), with *G. zealandicum* NRRL 22465 used as the outgroup. The resulting phylogenomic tree resolved into two well-supported major clades at the genus level (excluding the outgroup): *Fusarium* and *Neocosmospora*. Within *Fusarium*, four phylogenetically distinct subclades were identified, each representing a well-recognized species complex, viz., the FIESC, FHSC, FTSC, and FLSC. The species described in this study were distributed among these clades as follows: *F. qiannanense* KUNCC 3417 was placed within the FHSC; *F. puerense* KUNCC 3505 T and *F. dracaenophilum* KUNCC 3495 T within the FIESC; *F. wenshanense* KUNCC 3512 T within the FLSC; and *N. fungicola* KUNCC 11079 T, *N. alboflava* KUNCC 3509 T, and *N. solani* KUNCC 3556 within the *Neocosmospora* clade. The genome-based phylogenetic tree exhibited a topology highly consistent with that of the multilocus tree (Figure 1, Figure 2, Figure 3, Figure 4 and Figure 6).

### 3.4. Taxonomy

*Fusarium dracaenophilum* Y.B. Wang, Q. Fan & Zhu L. Yang, sp. nov.

Fungal Names No.: FN 572883

(Figure 7)

Etymology: The epithet “*dracaenophilum*” refers to the fungus’s ecological association with its host plant, *Dracaena cambodiana* Pierre ex Gagnep.

Type: CHINA, Yunnan Province, Xishuangbanna Dai Autonomous Prefecture, 100.77° E, 22° N, alt. 550 m, from the healthy leaves of *D. cambodiana*, June 2022, Z.Y. Tian (holotype HKAS 135090, ex-type culture KUNCC 3495).

Description: Conidiophores arising from aerial mycelium 7.9–16.0 μm tall, simple or rarely irregularly branched, bearing whorls of 2–3 phialides at the apex. Aerial conidiogenous cells monophialidic, sometimes reduced to solitary forms and laterally borne on hyphae, subulate to subcylindrical, smooth- and thin-walled, (7.7–)8.0–17.4(–18.6) × (2.8–)2.9–3.9(–4.0) μm (av. 12.1 × 3.4 μm), apical collarettes absent and periclinal thickening inconspicuous. Aerial macroconidia falcate to navicular, hyaline, smooth- and thin-walled, almost straight to slightly dorsiventrally curved, apical cell blunt to slightly curved, basal cell stunted to well-developed, foot-shaped, (1–)3–7-septate, predominantly 5-septate; 1-septate conidia: 16.0–27.0(–31.0) × 2.0–4.0 μm (av. 19.6 × 3.0 μm, *n* = 10); 2-septate conidia: (19.5–)21.5–30.0(–33.0) × 3.0–4.5 μm (av. 25.0 × 4.0 μm, *n* = 7); 3-septate conidia: (25.0–)26.5–37.5 × (3.0–)3.0–4.5(–5.0) μm (av. 30.5 × 4.0 μm, *n* = 20); 4-septate conidia: (30.0–)31.5–40.0(–41.5) × 4.0–5.0 μm (av. 35.5 × 4.0 μm, *n* = 20); 5-septate conidia: (31.0–)33.3–54.1(–62.0) × 3.5–5.5(–6.0) μm (av. 44.0 × 4.5 μm, *n* = 30); 6-septate conidia: 51.0–65.5(–67.0) × 3.5–6.0 μm (av. 58.0 × 4.0 μm, *n* = 20); 7-septate conidia: (50.5–)52.5–71.0(–74.0) × 4.0–5.5 μm (av. 60.5 × 4.5 μm, *n* = 20); overall: 16.0–71.0(–74.0) × 2.0–6.0 μm (av. 40.0 × 4.0 μm, *n* = 127). Sporodochia and chlamydospores not observed.

Culture characteristics: Colonies on PDA grow rapidly, exhibiting 5.5–6.0 cm diam. in seven days at 25 °C, moist, flat, aerial mycelium absent, colony margin regular, surface white to cream, reverse pale white to cream. On SNA reaching 6.5–7.0 cm diam. in seven days, moist, aerial mycelium absent, colony margin regular, surface white to cream, reverse pale white to cream. On OA reaching 6.0–6.5 cm diam. in seven days, flat, aerial mycelium absent, moist at the center, velvety at the margin, colony margin regular, surface white to cream, reverse white to cream.

Additional specimens examined: CHINA, Yunnan Province, Xishuangbanna Dai Autonomous Prefecture, 100.77° E, 22° N, alt. 550 m, from the healthy leaves of *D. cambodiana*, June 2022, Z.Y. Tian (culture KUNCC 3504).

Note: Phylogenetic analyses based on the concatenated dataset of *CaM*, *tef1*, *rpb1*, and *rpb2* loci (Figure 2) and genomic datasets (Figure 6) resolved the isolates representing *F. dracaenophilum* as a strongly supported monophyletic clade within the FIESC (BS = 100%, PP = 1.00 for multigene phylogenetic trees, BS = 100% for phylogenomic tree). *Fusarium dracaenophilum* is closely related to *F*. *weifangense*, *F*. *caulendophyticum*, and *F. citri* but can be distinguished by sequence differences of 28 bp, 36 bp, and 71 bp in the three-locus dataset, respectively. Morphologically, *F. dracaenophilum* can be distinguished from related species by its larger macroconidia (16.0–74.0 × 2.0–6.0 μm in *F. dracaenophilum* vs. 26.5–49.4 × 4.1–7.1 μm in *F. weifangense*, 5.0–40.5 × 3.0–5.5 μm in *F. citri*, and 10.0–47.0 × 2.0–4.5 μm in *F. caulendophyticum*) and more septation (1–7-septate in *F. dracaenophilum* vs. 3–7-septate in *F. weifangense*, 3–5-septate in *F. caulendophyticum*, and 1–5-septate in *F. citri*) [5,6,61]. Furthermore, the PHI test indicated no significant recombination (*P* = 0.736) between *F. dracaenophilum* and its closely related taxa (Figure 5A). Thus, *F. dracaenophilum* is hereby described as a new species within the FIESC.

*Fusarium puerense* Y.B. Wang, Q. Fan & Zhu L. Yang, sp. nov.

Fungal Names No.: FN 572884

(Figure 8)

Etymology: Named after the city, Pu’er, where the holotype was collected.

*Type:* CHINA, Yunnan Province, Pu’er city, 101.51° E, 23.35° N, alt. 709 m, from the symptomatic tissues of *Musa* sp., July 2023, Q. Fan (holotype HKAS 135091, ex-type culture KUNCC 3505).

Description: Aerial conidiophores and conidia were not detected during observation. Sporodochia cream to yellowish, translucent, formed densely on carnation leaves and on the agar. Sporodochial conidiophores densely, irregularly branched, 8.0–13.0 × 3.0–4.0 μm, bearing apical whorls of 1–2 phialides. Sporodochial conidiogenous cells monophialidic, flask-shaped, 9.0–15.5 × 2.5–4.5 μm, smooth- and thin-walled, apical collarettes absent and periclinal thickening inconspicuous. Sporodochial microconidia absent. Sporodochial macroconidia falcate, hyaline, smooth- and thin-walled, straight to slightly dorsiventrally curved, broadest at the middle portion, tapering towards both ends, apical cell blunt to slightly curved, basal cell stunted to well-developed, foot-shaped, 3–7-septate, predominantly 5-septate; 3-septate conidia: 23.0–40.5 × 3.5–5.0 μm (av. 32.0 × 4.0 μm, *n* = 10); 4-septate conidia: 35.0–69.0(–73.5) × 3.5–5.0 μm (av. 44.0 × 4.0 μm, *n* = 15); 5-septate conidia: (54.0–)56.0–78.0 × 4.0–5.0 μm (av. 67.5 × 4.5 μm, *n* = 20); 6-septate conidia: (64.0–)67.0–80.0 × 4.5–5.0 μm (av. 76.0 × 4.5 μm, *n* = 15); 7-septate conidia: 76.0–88.0 × 4.0–5.0 μm (av. 82.0 × 4.5 μm, *n* = 10); overall: 23.0–88.0 × 4.0–5.0 μm (av. 60.0 × 4.0 μm, *n* = 70). Chlamydospores not observed.

*Culture characteristics*: Colonies on PDA grow rapidly, exhibiting 4.5–5.5 cm diam. in seven days at 25 °C, cottony, flat, aerial mycelium abundant, colony margin regular, surface white, reverse white to cream. On SNA reaching 6.5–7.0 cm diam. in seven days, flat, aerial mycelium scant, moist, colony margin regular; surface white, reverse white to cream. On OA reaching 5.5–6.0 cm diam. in seven days, dense, with abundant aerial mycelium, surface pale orange in the center, white at the margin, reverse pale orange.

Additional specimens examined: CHINA, Yunnan Province, Xishuangbanna Dai Autonomous Prefecture, 100.77° E, 22° N, alt. 550 m, from the healthy leaves of *D. cambodiana*, June 2022, Z.Y. Tian (cultures: KUNCC 3501, KUNCC 3502, and KUNCC 3503).

Note: Phylogenetic analysis based on the concatenated dataset of *CaM*, *tef1*, *rpb1*, and *rpb2* loci (Figure 2) and genomic datasets (Figure 6) resolved the isolates representing *F. puerense* as a monophyletic clade within the FIESC lineage, with strong statistical support (BS = 100%, PP = 1.00 for multigene phylogenetic trees, BS = 100% for phylogenomic tree). *Fusarium puerense* is closely related to *F*. *ipomoeae* and *F*. *caulicola* but can be distinguished by sequence differences of 43 bp and 20 bp in the combined dataset, respectively. Morphologically, *F. puerense* differs from related species in macroconidial size (23.0–88.0 × 4.0–5.0 μm in *F. puerense* vs. 26.5–57.0 × 3.0–5.0 μm in *F. ipomoeae* and 14.0–40.5 × 2.0–5.0 μm in *F. caulicola*) and septation (3–7-septate in *F. puerense* vs. 1–4-septate in *F. caulicola*, 3–5-septate in *F. ipomoeae*) [5,61,62]. Furthermore, the PHI test indicated no significant recombination (*P* = 0.468) between *F. puerense* and its closely related taxa (Figure 5B). Thus, *F. puerense* is introduced as a new species.

*Fusarium wenshanense* Y.B. Wang, Q. Fan & Zhu L. Yang, sp. nov.

Fungal Names No.: FN 572885

(Figure 9)

Etymology: Named after the city, Wenshan Zhuang and Miao Autonomous Prefecture, where the holotype was collected.

*Type:* CHINA, Yunnan Province, Wenshan Zhuang and Miao Autonomous Prefecture, 103.87° E, 23.61° N, alt. 1810 m, from adult of *Lepidoptera*, July 2023, C.Y. Wei (holotype HKAS 135098, ex-type culture KUNCC 3512).

Description: Aerial conidiophores and conidia were not detected during observation. Sporodochia pale orange to orange, translucent, sparsely formed on carnation leaves and on the agar, often covered with aerial mycelium. Sporodochial conidiophores densely, irregularly branched, 16.2–19.3 × 2.6–3.8 μm, bearing apical whorls of 2–3 phialides, rarely solitary phialides. Sporodochial conidiogenous cells monophialidic, subulate to subcylindrical, (16.5–)18.0–26.5(–28.0) × 1.5–3.0 μm, smooth and thin-walled, apical collarettes absent and periclinal thickening inconspicuous. Sporodochial microconidia absent. Sporodochial macroconidia falcate, hyaline, smooth- and thin-walled, straight to slightly dorsiventrally curved, tapering towards both ends, apical cell blunt to papillate, basal cell stunted to well-developed, foot-shaped, (1–)3–7(–8)-septate, predominantly 5-septate; 1-septate conidia: 14.0–17.0(–18.0) × (3.0–)4.0–4.5 μm (av. 15.0 × 4.0 μm, *n* = 8); 2-septate conidia: (24.0–)25.5–32.5(–35.0) × (3.0–)4.0–4.5 μm (av. 29.5 × 4.0 μm, *n* = 6); 3-septate conidia: 35.5–49.0 × 4.0–4.5 μm (av. 42.0 × 4.0 μm, *n* = 15); 4-septate conidia: 44.0–60.0 × 3.5–5.0 μm (av. 52.0 × 4.0 μm, *n* = 15); 5-septate conidia: (48.0–)52.0–69.0 ×4.0–5.5 μm (av. 62.0 × 5.0 μm, *n* =30); 6-septate conidia: (60.0–)61.5–75.6 × 4.0–5.0 μm (av. 69.5 × 5.0 μm, *n* = 15); 7-septate conidia: 73.5–85.0(–87.0) ×4.5–5.5 μm (av. 78.0 × 5.0 μm, *n* = 15); 8-septate conidia: (77.0–)81–89(–92.5) × 4.0–5.0 μm (av. 85 × 4.5 μm, *n* = 6); overall: 14.0–89(–92.5) × (3.0–)4.0–5.0 μm (av. 54.0 × 4.5 μm) (*n* = 110). Chlamydospores obovoidal, subglobose to globose, hyaline, smooth-walled to slightly roughened, thick-walled, 9.5–18.8 μm, terminal or intercalary, solitary, in pairs or forming chains.

Colonies on PDA exhibiting 3.5–4.0 cm diam. in seven days at 25 °C, aerial mycelium abundant, dense, flat, colony margin regular, surface white to cream, reverse cream. On SNA reaching 6.5–7.0 cm diam. in seven days, aerial mycelium scant, flat, colony margin regular; surface white to cream, reverse white to cream. On OA reaching 4.5–5.0 cm diam. in seven days, moist at center, abundant aerial mycelium at margin, dense, colony margin regular, surface white, reverse white.

Additional specimens examined: CHINA, Yunnan Province, Wenshan Zhuang and Miao Autonomous Prefecture, 103.87° E, 23.61° N, alt. 1810 m, from adult of *Lepidoptera*, July 2023, C.Y. Wei (cultures: KUNCC 3510 and KUNCC 3511).

Note: Phylogenetic analysis based on the concatenated dataset of *CaM*, *tef1*, *rpb2*, and *tub2* loci (Figure 3) and genomic datasets (Figure 6) resolved the representing isolates of *F. wenshanense* as a monophyletic clade within the FLSC lineage, with strong statistical support (BS = 100%, PP = 1.00 for multigene phylogenetic trees, BS = 100% for phylogenomic tree). *Fusarium wenshanense* is closely related to *F. citri-sinensis* and *F. fujianense* but differs by 25 bp from *F. citri-sinensis* in the 4-locus (*CaM*-*tef1*-*rpb2*-*tub2*) dataset and 23 bp from *F. fujianense* in the 2-locus (*tef1*-*rpb2*) dataset (*CaM* and *tub2* sequences are not available for *F. fujianense*). Morphologically, *F. wenshanense* can be distinguished from related species by its sporodochial macroconidial size (14.0–92.5 × 3.0–5.0 μm in *F. wenshanense* vs. 39.7–99.5 × 4.0–7.7 μm in *F. citri-sinensis*, and 40.2–63.4 × 4.5–6.9 μm in *F. fujianense*) and septation (1–8-septate in *F. wenshanense* vs. 3–13-septate in *F. citri-sinensis*, and 4–6-septate in *F. fujianense*) [77,78]. Furthermore, the PHI test indicated no significant recombination (*P* = 1.0) between *F. wenshanense* and its closely related taxa (Figure 5C). Thus, *F. wenshanense* is introduced as a new species.

*Fusarium qiannanense* H. Zhang & Y.L. Jiang, Mycosphere 14(1): 2105 (2023)

Index Fungorum No.: IF900486

(Figure 10)

Description: Aerial conidiophores were not detected during observation. Aerial conidiogenous cells monophialidic, often reduced to solitary cells laterally borne on hyphae, subulate to subcylindrical, smooth- and thin-walled, 7.0–36.5 × 2.0–5.0 μm (av. 20.5 × 3.5 μm), apical collarettes absent and periclinal thickening inconspicuous. Aerial conidia of two types: microconidia oval to broadly ellipsoidal, straight to slightly curved, hyaline, smooth- and thin-walled, 0(–1)-septate, 0-septate conidia: (8.0–)10.0–18.0(–21.0) × 3.0–5.0(–6.0) μm (av. 14.0 × 4.0 μm, *n* = 26); 1-septate conidia: (11–)13–18 × 3.0–3.5 μm (av. 16.0 × 4.0, μm, *n* = 8); macroconidia falcate to navicular, hyaline, smooth- and thin-walled, almost straight to slightly dorsiventrally curved, apical cell blunt or papillate, basal cell stunted to well-developed, foot-shaped, 1–3(–5)-septate, predominantly 1-septate, 1-septate conidia: (14.0–)15.0–26.5(–27.0) × 3.0–4.0 μm (av. 21.0 × 3.5 μm, *n* = 21); 2-septate conidia: (26–)27.0–37.0(–38.5) × 3.0–4.5 μm (av. 31.5 × 3.5 μm, *n* = 15); 3-septate conidia: (17.0–)24.0–36.0(–38) × 3.5–4.5 μm (av. 28.5 × 4.0 μm, *n* = 10); 5-septate conidia: 35.0–48.5 × 4.0–4.5 μm (av. 38.0 × 4.0 μm, *n* = 4); overall: (14.0–)15.0–48.5 × 3.0–4.5 μm (av. 30.0 × 4.0 μm) (*n* = 50). Sporodochia and chlamydospores not observed.

Culture characteristics: Colonies on PDA exhibiting 4.5–5.0 cm diam. in seven days at 25 °C, velvety, flat, with abundant aerial mycelium, colony margin regular; surface white to cream, reverse cream. On SNA reaching 3.0–3.5 cm diam. in seven days, flat, aerial mycelium abundant, colony margin regular, surface white, reverse white. On OA reaching 5.0–6.0 cm diam. in seven days, flat, aerial mycelium scant, colony margin regular; surface white to cream, reverse white to cream.

Additional specimens examined: CHINA, Yunnan Province, Kunming city, 103.12° E, 25.31° N, alt. 1543 m, from the asymptomatic sclerotium of *Claviceps purpurea*, September 2023, M.L. Ding (cultures: KUNCC 3417, KUNCC 3416, and KUNCC 3415).

Note: Phylogenetic analyses revealed that the isolates KUNCC 3417, KUNCC 3416, and KUNCC 3415 clustered together with the type strain CGMCC 3.25477 of *F. qiannanense* with strong statistical support (BS = 100%, PP = 1.00) (Figure 1). The strains showed high sequence similarity across ITS, *tef1*, *rpb1*, and *rpb2* regions, showing 99.22% (512/516, 4 gaps), 100% (582/582, no gaps), 99.89% (1739/1741, 2 gaps), and 99.89% (911/912, 1 gap) identity, respectively. Morphologically, these isolates were characterized by monophialidic conidiogenous cells and straight to slightly curved aerial macroconidia. Both molecular and morphological evidence supported the identification of these isolates as *F. qiannanense*. As *F. qiannanense* was previously reported only from *Rosa roxburghii* (Rosaceae), the present study represents a new host record from *C. purpurea* [5].

*Neocosmospora alboflava* Y.B. Wang, Q. Fan & Zhu L. Yang, sp. nov.

Fungal Names No.: FN 572886

(Figure 11)

Etymology: Referring to its colony color, it develops white and pale orange interwoven concentric rings on PDA, SNA, and OA media.

*Type:* CHINA, Yunnan Province, Wenshan Zhuang and Miao Autonomous Prefecture, 104.39° N, 23.01° E, alt. 1699 m, from asymptomatic *Nigelia* sp., June 2023, C.Y. Wei (holotype HKAS 135095, ex-type culture KUNCC 3509).

Description: Conidiophores arising from aerial mycelium 20.8–62.3 μm tall, simple to rarely irregularly branched, bearing terminal single phialides or whorls of 2–3 phialides, commonly reduced to solitary phialides borne laterally on hyphae. Aerial conidiogenous cells monophialidic, subulate to subcylindrical, smooth- and thin-walled, (24.5–)36.5–87.0(–93.0) × 2.0–3.0 μm (av. 50.0 × 3.0 μm, *n* = 28), apical collarettes absent and periclinal thickening inconspicuous. Aerial microconidia arranged in false heads on phialide tips, obovoid to short clavate, straight to slightly curved, hyaline, smooth- and thin-walled, aseptate: 5.0–10.5 × 2.5–5.5 μm (av. 7.0 × 4.0 μm, *n* = 97). Chlamydospores, subglobose to globose, hyaline to pale yellow brown, smooth-walled to slightly roughened, thick-walled, 7.5–11.0 μm, commonly intercalary, in pairs or forming chains.

Culture characteristics: Colonies on PDA grow rapidly, exhibiting 6.5–7.0 cm diam. in seven days at 25 °C, velvety, flat, aerial mycelium abundant, exhibiting white and pale orange interwoven concentric rings, surface pale orange at margin, white to cream at center, reverse cream to pale orange, colony margin regular. On SNA reaching 6.0–6.5 cm diam. in seven days, flat, exhibiting white and pale orange interwoven concentric rings, surface pale orange at margin, white to cream at center, reverse cream, colony margin regular. On OA reaching 4.5–5.0 cm diam. in seven days, dense, exhibiting white and pale orange interwoven concentric rings, surface pale orange at margin, white to cream at center, reverse cream to pale orange, colony margin regular.

Additional specimens examined: CHINA, Yunnan Province, Wenshan Zhuang and Miao Autonomous Prefecture, 104.39° N, 23.01° E, alt. 1699 m, from asymptomatic *Nigelia* sp., June 2023, C.Y. Wei (cultures: KUNCC 3526, KUNCC 3527, and KUNCC 3528).

Note: Phylogenetic analysis based on the concatenated dataset of ITS, *CaM*, *acl1*, *tef1*, *rpb1*, and *rpb2* loci (Figure 4) and genomic dataset (Figure 6) resolved the representative isolates of *N. alboflava* as a well-supported monophyletic clade within *Neocosmospora* (BS = 80%, PP = 0.80 for multilocus phylogenetic trees, BS = 100% for phylogenomic tree). *Neocosmospora alboflava* is closely related to *N. parceramosa*, *N. liriodendri*, and *N. pseudoradicicola*, but differs by sequence differences of 34 bp and 55 bp from *N. liriodendri* and *N. pseudoradicicola* in the 6-locus (ITS-*CaM*-*acl1-tef1-rpb1-rpb2*) dataset, respectively, and differs by 54 bp from *N. parceramosa* in the 5-locus (ITS-*CaM-acl1-tef1-rpb2*) dataset (*rpb1* sequence is not available for *N. parceramosa*). Morphologically, *N. alboflava* can be distinguished from related species based on phialides size (24.5–93.0 × 2.0–3.0 μm in *N. alboflava* vs. 39.5–78 × 2–4.5 μm in *N. pseudoradicicola*, 40–71.5 × 2.5–5 μm in *N. liriodendri*, and 35–74 × 2–4 μm in *N. parceramosa*) and microconidia septation (0-septate in *N. alboflava* vs. 0(–1)-septate in *N. parceramosa*, *N. liriodendri*, and *N. pseudoradicicola*) [8]. In addition, the PHI test detected no significant recombination (*P* = 0.887) between *N. alboflava* and its closely related taxa (Figure 5D). Thus, *N. alboflava* is introduced as a new species.

*Neocosmospora fungicola* Y.B. Wang, Q. Fan & Zhu L. Yang, sp. nov.

Fungal Names No.: FN 572887

(Figure 12)

Etymology: Named after its isolation from the fungus.

Type: CHINA, Zhejiang Province, Hangzhou City, 120.27° E, 30.25° N, alt. 90 m, isolated from the asymptomatic *Ophiocordyceps* sp., June 2022, L.Y. Xie (holotype HKAS 126202, ex-type culture KUNCC 11079).

Description: Conidiophores on the aerial mycelium straight or flexuous, smooth- and thin-walled, mostly simple or irregularly branched, bearing phialides dichotomously at the apex, rarely solitary phialides. Aerial conidiogenous cells monophialidic, sometimes reduced to solitary phialides borne laterally on hyphae, subcylindrical, smooth- and thin-walled, 41.0–62.0(–65.5) × 2.0–3.0 μm (av. 49.0 × 2.0 μm), apical collarettes absent and periclinal thickening inconspicuous. Aerial conidia obovoid to short clavate, straight to slightly curved, hyaline, smooth- and thin-walled, 0–1-septate, 0-septate conidia: 5.5–10.0(–12) × 2.0–5.0 μm (av. 8.0 × 3.0 μm, *n* = 33); 1-septate conidia: 9.0–12.0 × 3.0–4.0 μm (av. 10.0 × 3.5 μm, *n* =14). Sporodochia pale honey, translucent, formed sparsely on carnation leaves and on the agar. Sporodochial conidiophores densely, irregularly branched, bearing apical whorls of 3–5 phialides. Sporodochial conidiogenous cells monophialidic, subulate to subcylindrical, 14.0–19.0 × 2.5–4.5 μm (av. 16.0 × 3.5 μm), smooth and thin-walled, apical collarettes absent and periclinal thickening inconspicuous. Sporodochial macroconidia falcate, hyaline, smooth- and thin-walled, straight to slightly dorsiventrally curved, broadest at the middle portion, tapering towards both ends, apical cell blunt to slightly curved, basal cell stunted to well-developed, foot-shaped, (3–)4–6(–7)-septate, predominantly 5-septate; 3-septate conidia: (33.0–)45.1–48.0 ×4.0–4.5 μm (av. 43.0 × 4.5 μm, *n* = 10); 4-septate conidia: 44.0–56.0 × 4.0–5.0 μm (av. 50.0 × 5.0 μm, *n* = 18); 5-septate conidia: 49.0–60.0 × 4.0–6.0 μm (av. 54.5 × 5.0 μm, *n* = 20); 6-septate conidia: 54.0–61.0 × 4.0–5.0 μm (av. 59.0 × 4.5 μm, *n* = 18); 7-septate conidia: 58.0–65.0 × 4.5–5.0 μm (av. 61.5 × 5 μm, *n* = 9); overall: (33.0–)45.1–65.0 × 4.0–6.0 μm (av. 54.0 × 4.5 μm, *n* = 75). Chlamydospores not observed.

Culture characteristics: Colonies on PDA grow rapidly, exhibiting 7.0–8.0 cm diam. in seven days at 25 °C, velvety, flat, aerial mycelium abundant, colony margin regular, surface white to cream, reverse white to cream. On SNA reaching 7.0–8.0 cm diam. in seven days, velvety, flat, fluffy, colony margin regular, surface white to cream, reverse white to cream. On OA reaching 4.5–5.0 cm diam. in seven days, velvety, flat, moist at center, colony margin regular; surface white to cream, reverse white to cream.

Additional specimens examined: CHINA, Zhejiang Province, Hangzhou City, 120.27° E, 30.25° N, alt. 90 m, from the *Ophiocordyceps* sp., June 2022, L.Y. Xie (cultures: KUNCC 11080, KUNCC 11081, and KUNCC 11082).

Note: Phylogenetic analysis based on the concatenated dataset of ITS, *CaM*, *acl1*, *tef1*, *rpb1*, and *rpb2* loci (Figure 4) and genomic datasets (Figure 6) resolved the representing isolates of *N. fungicola* as a monophyletic clade within the *Neocosmospora*, which is statistically well supported (BS = 85%, PP = 1.00 for multigene phylogenetic trees, BS = 100% for phylogenomic tree). *Neocosmospora fungicola* is closely related to *N. parceramosa*, *N. liriodendri*, and *N. pseudoradicicola* but can be distinguished by sequence differences of 29 bp and 53 bp from *N. liriodendri* and *N. pseudoradicicola* in the 6-locus (ITS-*CaM-acl1-tef1-rpb1-rpb2*) dataset, respectively, and 16 bp from *N. parceramosa* in the 5-locus (ITS-*CaM-acl1-tef1-rpb2*) dataset (*rpb1* sequences are not available for *N. parceramosa*). Morphologically, *N. fungicola* can be distinguished from related species based on the size of phialides (24.5–93.0 × 2.0–3.0 μm in *N. alboflava* vs. 39.5–78 × 2–4.5 μm in *N. pseudoradicicola*, 40–71.5 × 2.5–5 μm in *N. liriodendri*, and 35–74 × 2–4 μm in *N. parceramosa*) and chlamydospores (chlamydospores absent in *N. alboflava* vs. chlamydospores present in *N. parceramosa*, *N. liriodendri*, and *N. pseudoradicicola*) [8]. Furthermore, the PHI test indicated no significant recombination (*P* = 0.887) between *N. fungicola* and its close relatives (Figure 5D). Thus, *N. fungicola* is introduced as a new species.

*Neocosmospora solani* (Mart.) L. Lombard & Crous, Stud. Mycol. 80: 228 (2015)

Index Fungorum No.: IF810964

(Figure 13)

Description: Sporodochia champagne, translucent, formed commonly on carnation leaves, rarely on aerial and substrate mycelium. Sporodochial conidiophores densely, irregularly branched, 10.0–15.0 × 3.5–5.0 μm, bearing apical whorls of 2–3 phialides, solitary phialides rarely. Sporodochial phialides monophialidic, doliiform to subcylindrical, 15.5–23.5 × 4.0–6.0 μm, smooth- and thin-walled, apical collarettes absent and periclinal thickening inconspicuous. Sporodochial microconidia absent. Sporodochial macroconidia lunate, hyaline, smooth- and thin-walled, straight to slightly dorsiventrally curved, broadest at the middle portion, tapering towards both ends, apical cell blunt to slightly curved, basal cell stunted to well-developed, foot-shaped, 1–4(5)-septate, predominantly 3-septate; 1-septate conidia: 20.5–26.5 × 4.0–6.5 μm (av. 23.0 × 5.5 μm, *n* = 15); 2-septate conidia: 25.5–29.0 × 5.0–5.5 μm (av. 27.0 × 5.4 μm, *n* = 12); 3-septate conidia: 38.5–44.0 × 6.0–7.5 μm (av. 42.0 × 7.0 μm, *n* = 32); 4-septate conidia: 39.5–48.0(–51.5) × 5.5–8.0 μm (av. 44.0 × 7.0 μm, *n* = 30); 5-septate conidia: 42.0–47.0 × 6.0–7.5 μm (av. 44.0 × 7.0 μm, *n* = 8). Chlamydospores not observed.

*Culture characteristics*: Colonies on PDA grow rapidly, exhibiting 7.0–7.5 cm diam. in seven days at 25 °C, cottony, flat, with abundant aerial mycelium, colony margin regular, surface white to cream, reverse white to cream. On SNA reaching 7.5–8.6 cm diam. in seven days, flat, aerial mycelium abundant, colony margin regular, surface white to cream, reverse cream. On OA reaching 4.5–5 cm diam. in seven days, dense, colony margin regular, surface white to cream, reverse cream to pale orange.

Additional specimens examined: CHINA, Yunnan Province, Xishuangbanna Dai Autonomous Prefecture, 100.77° E, 22° N, alt. 550 m, from the healthy leaves of *D. cambodiana*, June 2022, Z.Y. Tian (cultures: KUNCC 3556 and KUNCC 3557).

Note: Phylogenetic analyses demonstrated that the isolates KUNCC 3556 and KUNCC 3557 clustered with the ex-epitype strain CBS 140079 of *Neocosmospora solani* with strong statistical support (BS = 100%, PP = 1.00, Figure 4). Sequence similarity was very high across multiple loci, with ITS, *CaM*, *acl1*, *tef1*, *rpb1*, and *rpb2* showing 99.82% (558/559, 1 gap), 100% (562/562, no gaps), 99.66% (589/591, 2 gaps), 99.57% (693/696, 3 gaps), 100.00% (1312/1312, no gaps), and 99.64% (834/837, 3 gaps) identity, respectively. Morphologically, these isolates were similar, characterized by orange sporodochia, cylindrical to subcylindrical conidiogenous cells, and straight to slightly curved aerial macroconidia. Both molecular and morphological evidence support the identification of these isolates as *N. solani* [8]. Therefore, this study reports a new host record for this species isolated from *D. cambodiana*.

## 4. Discussion

### 4.1. Species Diversity of Fusarium and Neocosmospora

*Fusarium* and *Neocosmospora* are recognized as the most species-rich and ecologically diverse genera within the family *Nectriaceae* [3,30]. Currently, *Fusarium* includes over 420 accepted species [4,6,43], and *Neocosmospora* contains more than 140 species [3,25] (https://indexfungorum.org/Names/Names.asp, accessed on 28 May 2025).

Based on the concept of *Fusarium s. s.* (=*Gibberella*, also known as the “F3 clade”), this study integrated morphological characteristics, multilocus phylogenetic analyses, and genomic data to investigate the species diversity within fusarioid fungi. As a result, five new species from southeastern and southwestern China are described. In addition, this study also reports new host records for *F. qiannanense* and *N. solani*, isolated from *D. cambodiana* and the sclerotia of *C. purpurea*, respectively. While most species described in this study were isolated from plants or fungi, *F. wenshanense* was obtained from the cadaver of a lepidopteran insect. Nevertheless, its entomopathogenic potential remains unconfirmed due to a lack of direct evidence and experimental validation.

### 4.2. Diversity and Potential of Endophytic Fusarioid Fungi

Endophytic fungi are those that inhabit plant tissues without causing visible disease symptoms in their hosts [79]. Although members of *Fusarium* and *Neocosmospora* are widely recognized for their pathogenicity in plants [6,23,30,39,80,81], growing evidence indicates that their endophytic members have notable ecological and biotechnological potential, especially in natural product discovery and sustainable agriculture. For instance, endophytic *N. solani* has been reported to produce various bioactive secondary metabolites with anticancer, antimicrobial, and antioxidant activities, including jasmonates, camptothecin, and naphthoquinones [82]. Similarly, *F. oxysporum* strain Fo47, a well-characterized endophyte, has been employed as a biocontrol agent that induces host defense responses via the accumulation of salicylic acid (SA) and camalexin, thereby improving plant resistance to multiple pathogens [83].

Among isolates obtained in this study, all strains were identified as endophytes, with the exception of KUNCC 3505 (*F. puerense*), which was isolated from a diseased banana (*Musa* sp.), and three strains of *F. wenshanense* (KUNCC 3510, 3511, and 3512), whose ecological roles remain unclear. These findings suggest that the diversity of naturally occurring endophytic fusarioid fungi may be significantly underestimated, and numerous undescribed taxa likely remain to be discovered. This view is supported by a recent study conducted by Zhang et al. [5], which documented the diversity of endophytic *Fusarium* and related fungi in *Rosa roxburghii*. Notably, the known species *F. qiannanense*, originally isolated from healthy roots of *R. roxburghii* [5], was isolated for the first time from asymptomatic sclerotia of *C. purpurea*. Although there are currently no reports of pathogenicity associated with this species, its potential opportunistic behavior under host stress conditions warrants further investigation [84]. Moreover, the newly described species *F. dracaenophilum*, most strains of *F. puerense*, and *N. solani* were isolated from asymptomatic leaves of *D. cambodiana*. Previous studies have shown that *Fusarium* spp. are dominant colonizers of *Dracaena*, and experimental evidence has demonstrated that *F. proliferatum* can induce dragon’s blood secretion in this medicinal plant [85,86,87]. However, whether our isolates can promote dragon’s blood production remains unknown and requires further experimental validation. Remarkably, *N. fungicola* and *N. alboflava* represent the first records of endophytic colonization in entomopathogenic fungi, specifically in cordycipitoid fungi hosts. As reports of fusarioid fungi parasitizing insect-pathogenic fungi are extremely rare [88], our findings broaden the ecological niche and potential host spectrum of this important fungal group.

### 4.3. Evaluation of the Phylogenetic Resolution of Molecular Markers in Fusarium and Related Genera

Accurate species delimitation in *Fusarium* and its close relative *Neocosmospora* requires molecular markers with high phylogenetic resolution. Although the nuclear ribosomal ITS region has been widely used as a universal DNA barcode for fungi [89], several studies have shown that the ITS lacks sufficient discriminatory power within these genera and cannot effectively resolve closely related species [4,90]. In this study, single-locus phylogenetic analyses demonstrated that both *tef1* and *rpb2* exhibited strong phylogenetic informativeness across multiple species complexes. Notably, *tef1* achieved complete species-level resolution within both the FIESC and FLSC, which is highly consistent with previous findings [34,91,92]. Furthermore, several studies have suggested that *rpb2* offers greater resolution among closely related species [4,6], particularly within the FFSC, FIESC, and *F. sambucinum* species complex (FSAMSC). However, as observed in the current study, the species resolution of *rpb2* in the FIESC was slightly lower than that of *tef1*. This discrepancy may result from differences in the number of taxa analyzed or the underlying phylogenetic complexity.

### 4.4. Topological Congruence Between Genome-Scale and Multilocus Phylogenetic Trees

The genome-scale phylogenetic tree constructed in this study showed strong topological congruence with the multilocus phylogenetic tree. Both analyses clearly resolved *Fusarium* and *Neocosmospora* as two well-supported monophyletic lineages. The taxonomic positions of the species described herein were largely consistent across both trees, further validating their taxonomic placement within their respective species complexes. These results also provide indirect support for the reliability of traditional phylogenetic markers (particularly *tef1*, *rpb2*, and *rpb1*) for species delimitation within fusarioid fungi [4,31,60]. This is especially relevant in cases where genome-scale data are not readily available, underscoring the continued utility of these markers in fungal systematics.

Our findings expand the known species diversity of *Fusarium* and *Neocosmospora* and provide critical support for refined species delimitation [4,8]. Moreover, the newly described species exhibit congruence across morphological, phylogenetic, and genomic levels, further validating the effectiveness of the polyphasic taxonomic approach in identifying cryptic species [6].

## 5. Conclusions

This study integrated a polyphasic taxonomic approach using morphological characteristics, multilocus phylogeny, and phylogenomics to investigate the phylogenetic relationships and species diversity of *Fusarium* and *Neocosmospora*. Five novel species are described and two new host records are reported. The findings highlight the continued relevance of molecular markers such as *tef1* and *rpb2* in resolving relationships within fusarioid fungi. Phylogenetic trees based on both multilocus and genome-scale datasets produced highly congruent topologies, providing strong support for species delimitation and phylogenetic inference. These results clearly delineate the phylogenetic boundaries between *Fusarium* and *Neocosmospora*, supporting their recognition as distinct genera within the *Nectriaceae*. Furthermore, the isolation of strains from diverse ecological sources, including plants, insects, and fungi, underscores the underappreciated ecological diversity of fusarioid fungi.

Despite the aforementioned advancements, this study has several limitations. First, the limited number of isolates in certain lineages may restrict the resolution of intraspecific variation. Second, although genomic data were included, functional genomic analyses remain insufficient to fully elucidate pathogenicity factors and ecological adaptation mechanisms. Additionally, the geographic coverage of this study was relatively narrow, with certain habitats and host types underrepresented.

Future research should focus on broader and more systematic sampling across diverse regions and ecological contexts, coupled with transcriptomic and comparative genomic approaches, to further clarify the evolutionary history, host specificity, and pathogenic potential of this important fungal lineage.

## Figures and Tables

**Figure 1 biology-14-00871-f001:**
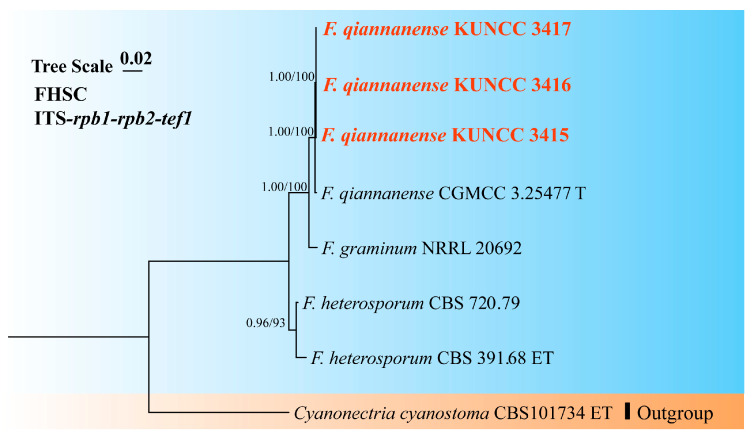
Multilocus phylogenetic tree of the *Fusarium heterosporum* species complex (FHSC). Strains newly isolated in this study are shown in red. Bayesian posterior probabilities (BI-PP > 0.9) and RAxML bootstrap support values (ML-BS > 70%) are indicated at the nodes (BI-PP/ML-BS). Ex-type and ex-epitype strains are shown in bold and marked with T and ET, respectively.

**Figure 2 biology-14-00871-f002:**
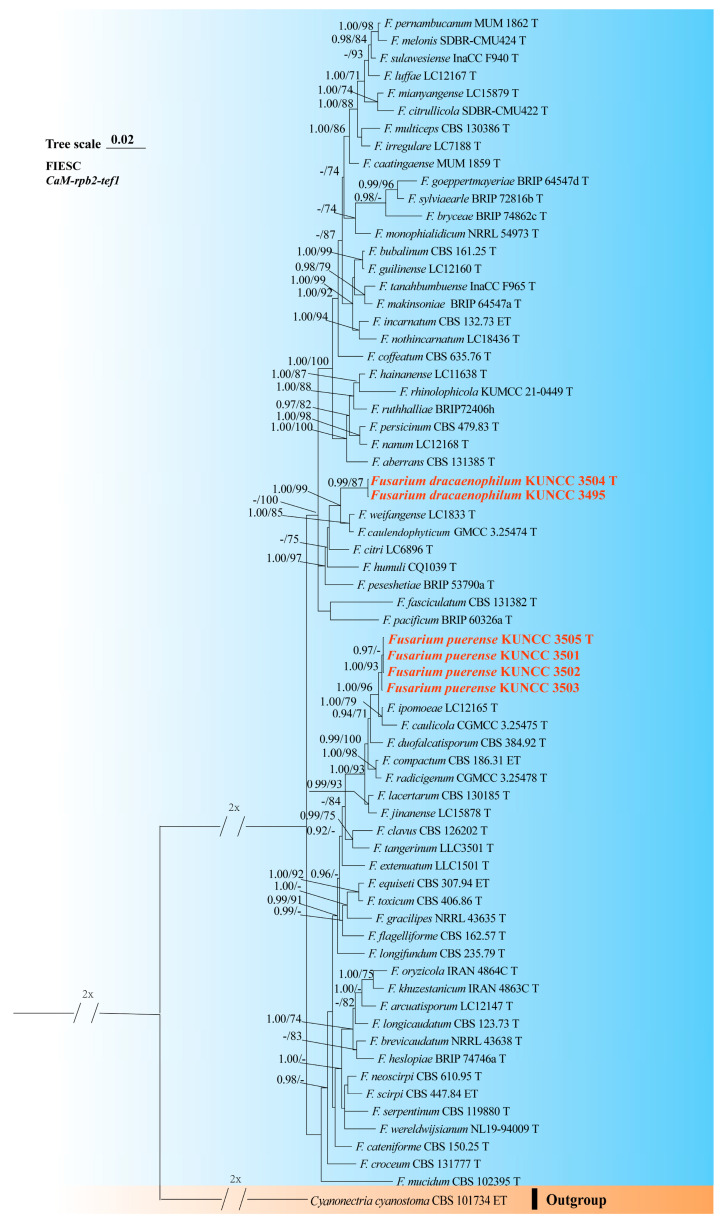
Multilocus phylogenetic tree of the *Fusarium incarnatum-equiseti* species complex (FIESC). Strains newly isolated in this study are shown in red. Bayesian posterior probabilities (BI-PP > 0.9) and RAxML bootstrap support values (ML-BS > 70%) are indicated at the nodes (BI-PP/ML-BS). Ex-type and ex-epitype strains are shown in bold and marked with T and ET, respectively.

**Figure 3 biology-14-00871-f003:**
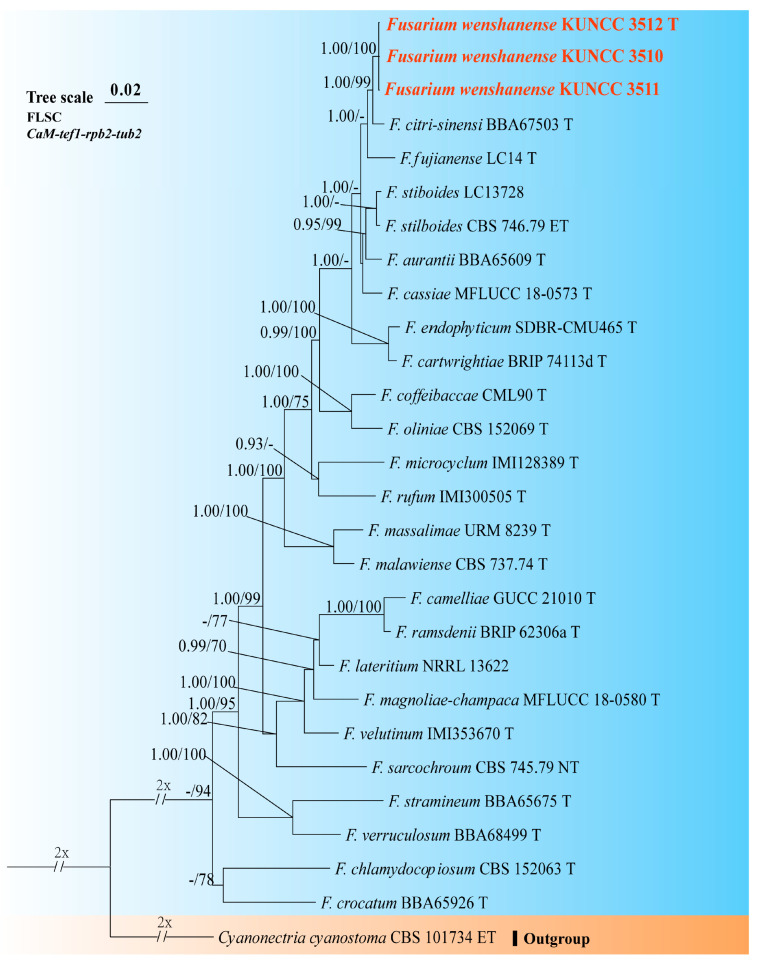
Multilocus phylogenetic tree of the *Fusarium lateritium* species complex (FLSC). Strains newly isolated in this study are shown in red. Bayesian posterior probabilities (BI-PP > 0.9) and RAxML bootstrap support values (ML-BS > 70%) are indicated at the nodes (BI-PP/ML-BS). Ex-type, ex-epitype, and ex-neotype strains are shown in bold and marked with T, ET, and NT, respectively.

**Figure 4 biology-14-00871-f004:**
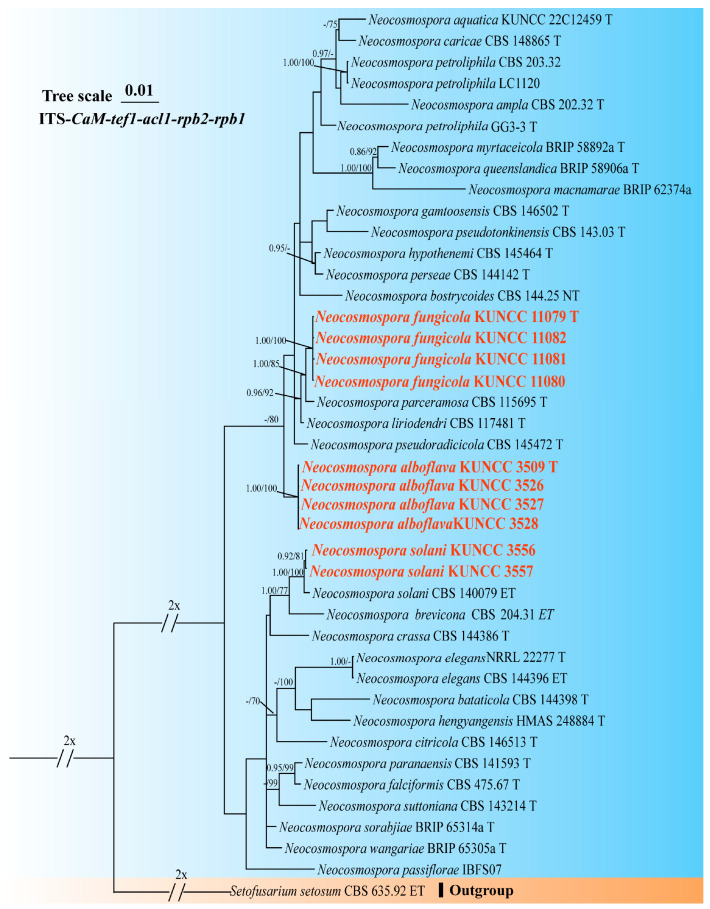
Multilocus phylogenetic tree of *Neocosmospora*. Strains newly isolated in this study are shown in red. Bayesian posterior probabilities (BI-PP > 0.9) and RAxML bootstrap support values (ML-BS > 70%) are indicated at the nodes (BI-PP/ML-BS). Ex-type, ex-epitype, and ex-neotype strains are shown in bold and marked with T, ET, and NT, respectively.

**Figure 5 biology-14-00871-f005:**
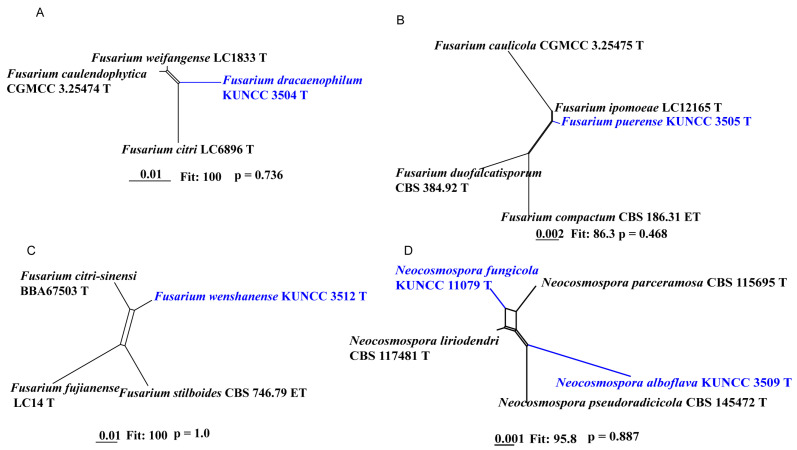
The pairwise homoplasy index (PHI) test of five new species and their closely related species. New taxa are printed in bold blue. (**A**) *Fusarium dracaenophilum* and its related species. (**B**) *F. puerense* and its related species. (**C**) *F. wenshanense* and its related species. (**D**) *Neocosmospora fungicola* and *N. alboflava* and their related species.

**Figure 6 biology-14-00871-f006:**
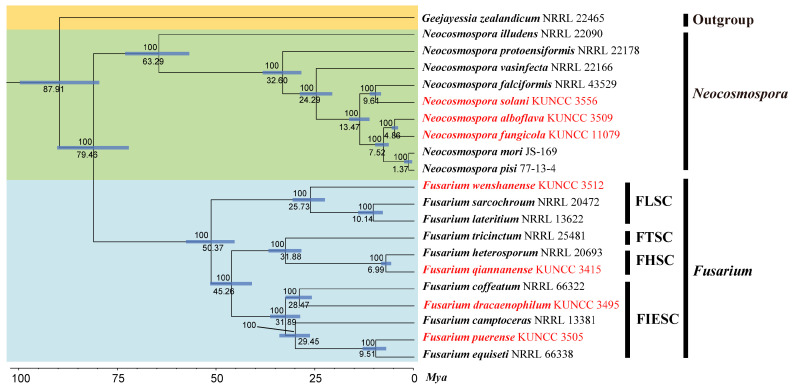
Phylogenomic tree of *Fusarium* and *Neocosmospora*. Node support values are shown as IQ-TREE ultrafast bootstrap values (UFBoot ≥ 95%). Blue bars indicate the 95% confidence intervals for divergence time estimates, with estimated divergence times (in million years ago, Mya) shown below each bar. Strains newly sequenced in this study are highlighted in red.

**Figure 7 biology-14-00871-f007:**
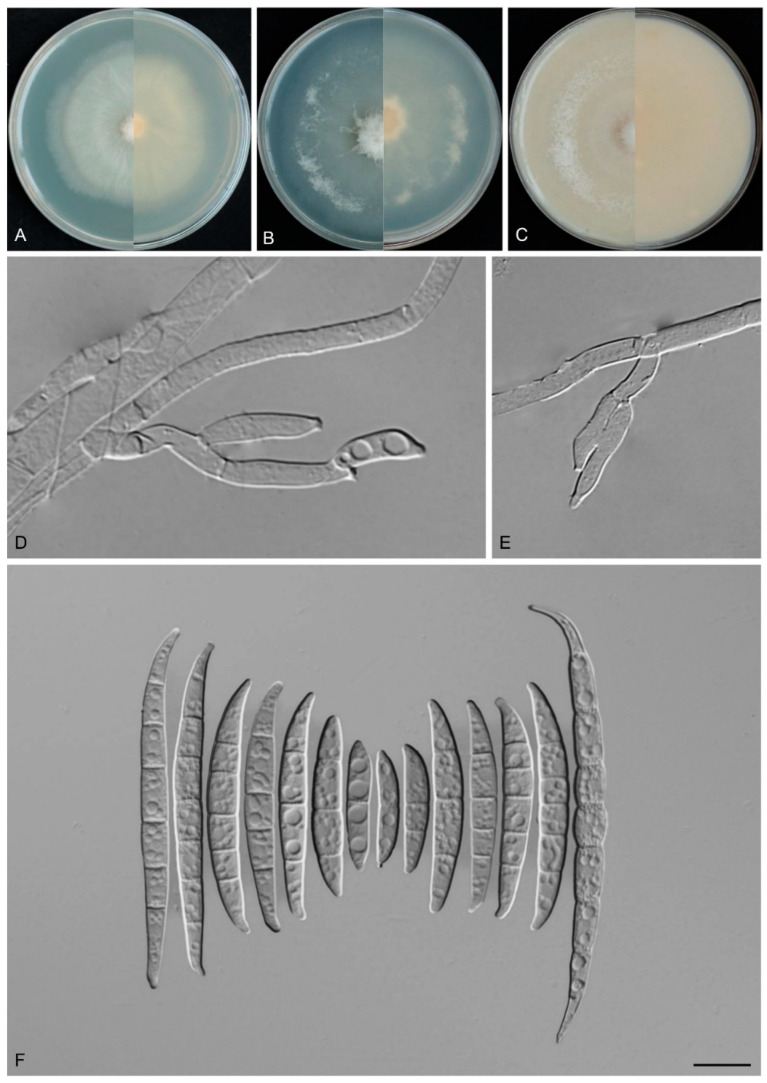
*Fusarium dracaenophilum* (ex-type culture KUNCC 3495). (**A**–**C**): Colony on PDA, SNA, and OA (**left**: surface; **right**: reverse). (**D**,**E**): Aerial conidiophores and conidiogenous cells. (**F**): Aerial macroconidia. Scale bars: (**D**–**F**) = 10 μm.

**Figure 8 biology-14-00871-f008:**
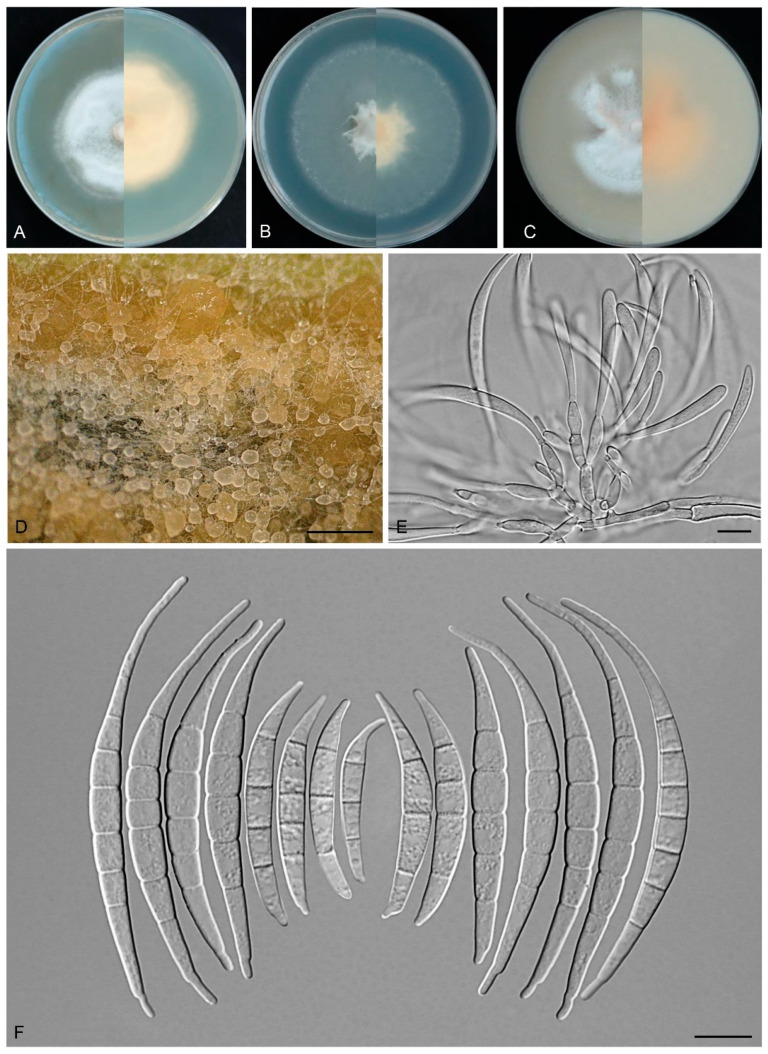
*Fusarium puerense* (ex-type culture KUNCC 3505). (**A**–**C**): Colony on PDA, SNA, and OA (**left**: surface; **right**: reverse). (**D**): Sporodochia. (**E**): Sporodochial conidiophores and conidiogenous cells. (**F**): Sporodochial macroconidia. Scale bars: (**D**) = 500 μm, (**E**,**F**) = 10 μm.

**Figure 9 biology-14-00871-f009:**
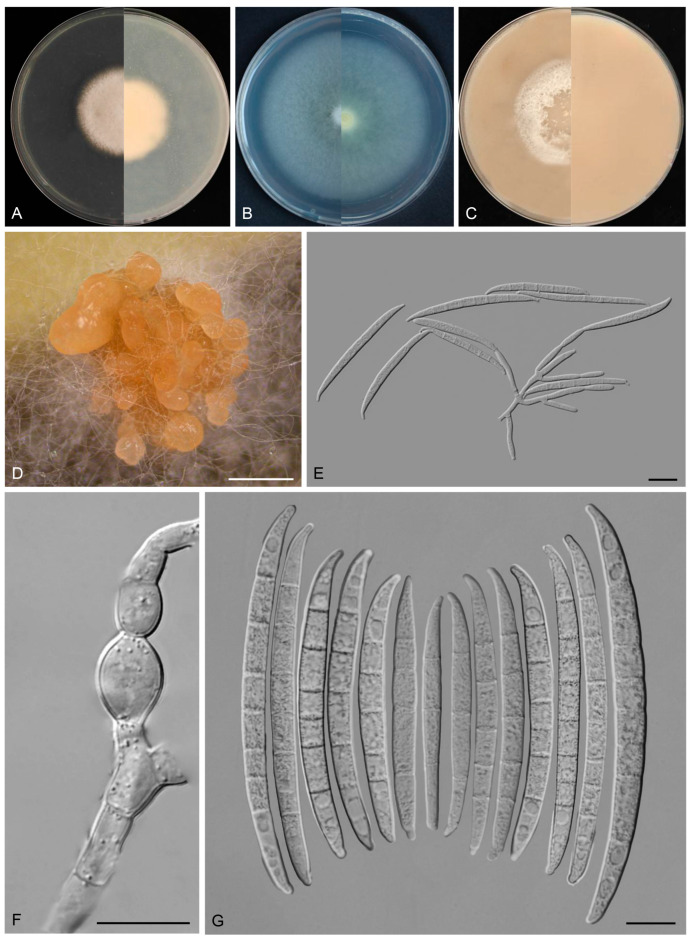
*Fusarium wenshanense* (ex-type culture KUNCC 3512). (**A**–**C**): Colony on PDA, SNA, and OA (**left**: surface; **right**: reverse). (**D**): Sporodochia. (**E**): Sporodochial conidiophores and conidiogenous cells. (**F**): Chlamydospores. (**G**): Sporodochial macroconidia. Scale bars: (**D**) = 500 μm, (**E**–**G**) = 10 μm.

**Figure 10 biology-14-00871-f010:**
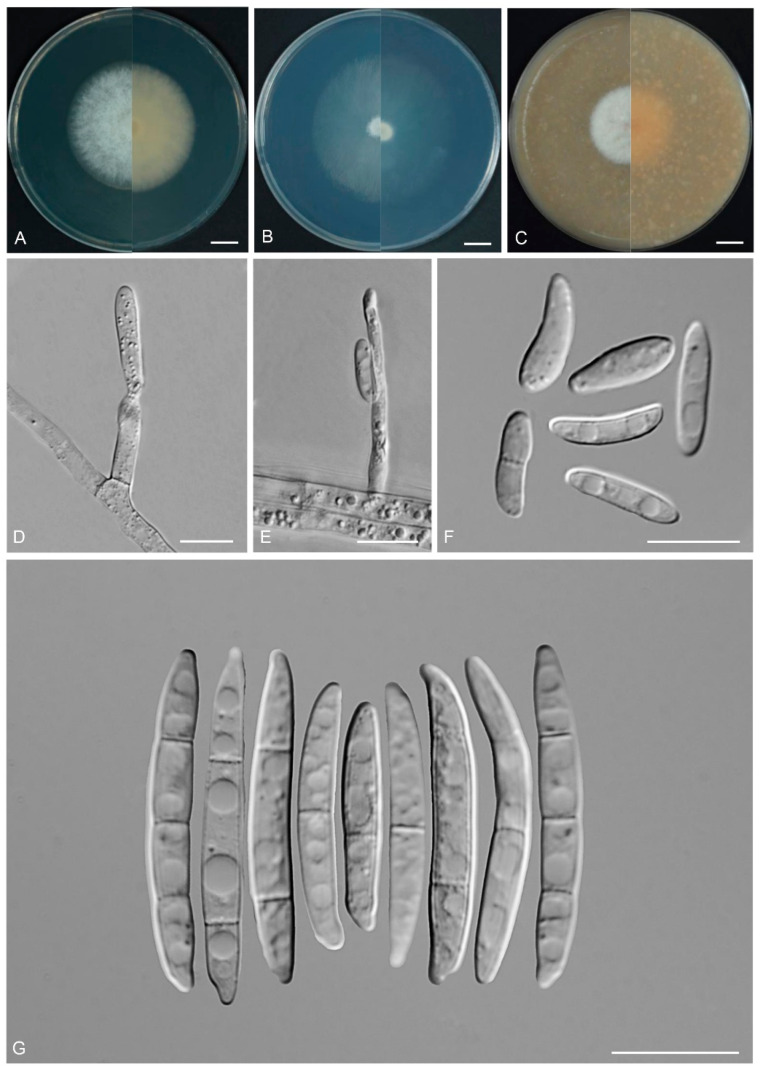
*Fusarium qiannanense* (culture KUNCC 3417). (**A**–**C**): Colony on PDA, SNA, and OA (**left**: surface; **right**: reverse). (**D**,**E**): Aerial conidiophores and conidiogenous cells. (**F**): Aerial microconidia. (**G**): Aerial macroconidia. Scale bars: (**D**–**G**) = 10 μm.

**Figure 11 biology-14-00871-f011:**
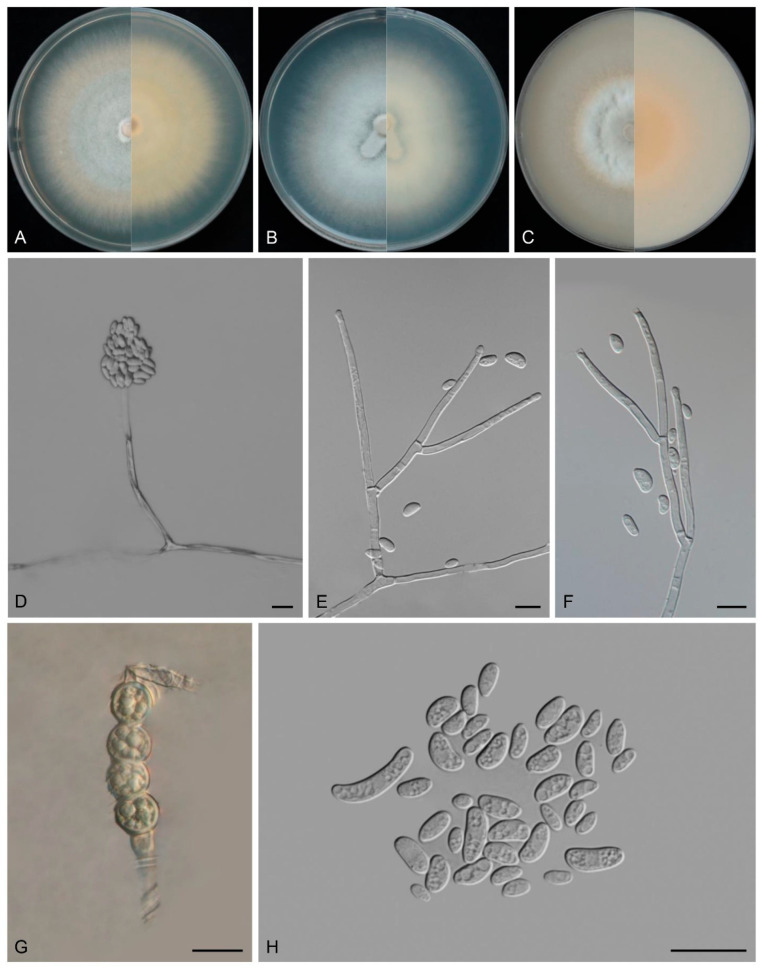
*Neocosmospora alboflava* (ex-type culture KUNCC 3509). (**A**–**C**): Colony on PDA, SNA, and OA (**left**: surface; **right**: reverse). (**D**–**F**): Aerial conidiophores and conidiogenous cells. (**G**): Chlamydospores. (**H**): Aerial microconidia. Scale bars: (**D**–**H**) = 10 μm.

**Figure 12 biology-14-00871-f012:**
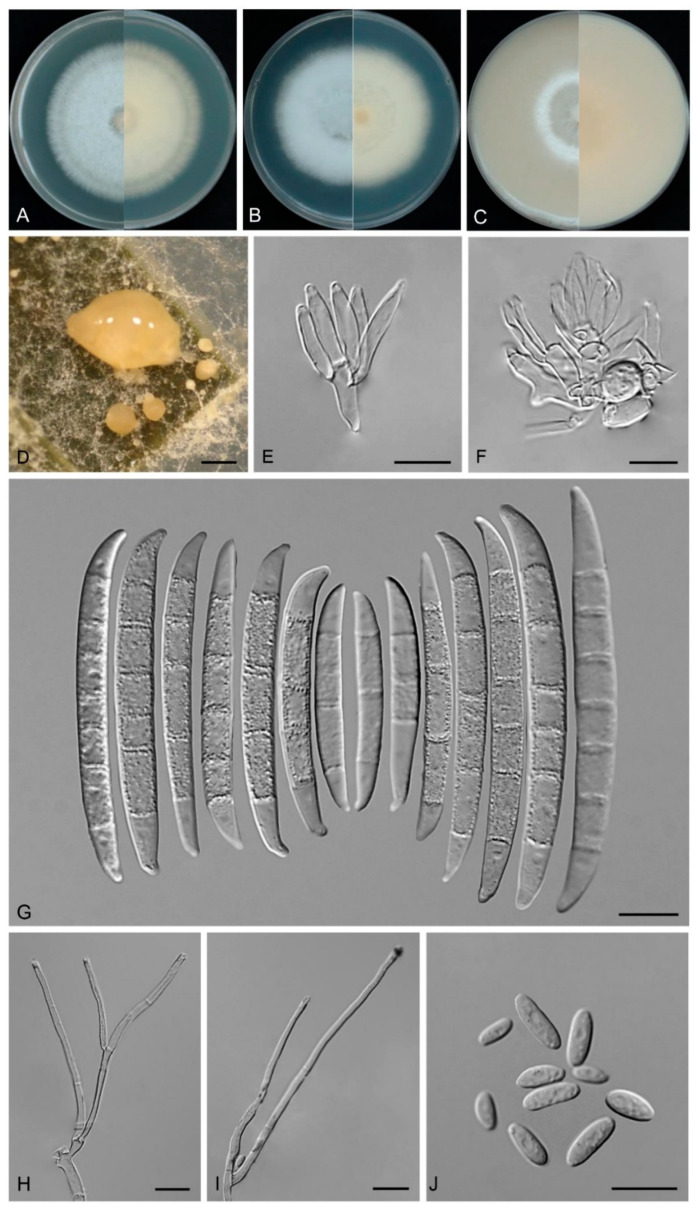
*Neocosmospora fungicola* (ex-type culture KUNCC 11079). (**A**–**C**): Colony on PDA, SNA, and OA (**left**: surface; **right**: reverse). (**D**): Sporodochia. (**E**,**F**): Sporodochial conidiophores and conidiogenous cells. (**G**): Sporodochial macroconidia. (**H**,**I**): Aerial conidiophores and conidiogenous cells. (**J**): Aerial microconidia. Scale bars: (**D**) = 500 μm, (**E**–**J**) = 10 μm.

**Figure 13 biology-14-00871-f013:**
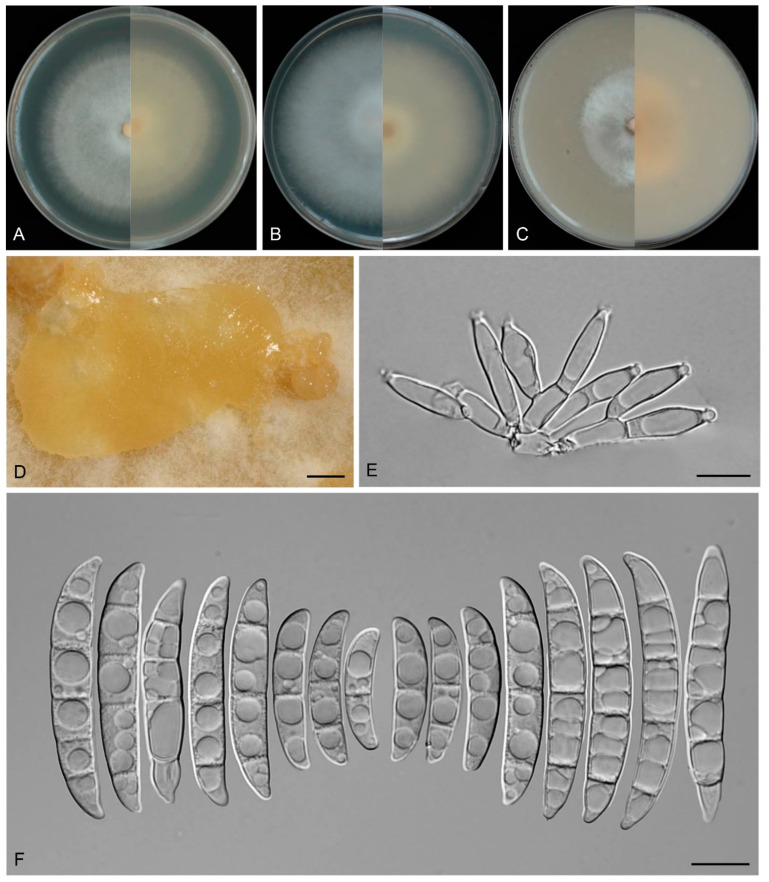
*Neocosmospora fungicola* (culture KUNCC 3556). (**A**–**C**): Colony on PDA, SNA, and OA (left: surface; right: reverse). (**D**): Sporodochia. (**E**): Sporodochial conidiophores and conidiogenous cells. (**F**): Sporodochial macroconidia. Scale bars: (**D**) = 500 μm, (**E**,**F**) = 10 μm.

## Data Availability

Genome assemblies were deposited in the National Center for Biotechnology Infommation (NCBI) Genome database under BioProject accession number PRJNA1247769 (https://www.ncbi.nlm.nih.gov/datasets/genome/, accessed on 11 April 2025). All data are available from the corresponding author upon request.

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
