# Peer review of "Phylogenomic, Morphological, and Phylogenetic Evidence Reveals Five New Species and Two New Host Records of Nectriaceae (Hypocreales) from China"

_biology, 2025, doi:10.3390/biology14070871_

Round 1

Reviewer 1 Report

Comments and Suggestions for Authors

I thank the authors for submit this very useful study in terms of identifying and revealing new genotypes of the studied Fusarium and Neocosmospora, which will help and benefit all researchers in the agricultural and medical fields. Nevertheless, please respond to the following comments, which aim to help improve the manuscript as much as possible.

  • In the introduction: lines 101-104, does the genotype vary by region, or do the authors mean the presence of new genotypes? On the other hand, the authors mention that in China, the characterization of Fusarium fungi is poor, but China is a very large country (where in China? is better to specify), Or it is better to specify an environmental area rather than mentioning the entire country.

  • In lines 108-109: Here, the purpose of the study and the strategy used should be specified, rather than stating the results as they currently exist. Please amend.

  • How was the purity of the fungal colonies determined after they were isolated from the three samples? The intent is to write the specifications of the pure fungal colonies to distinguish them from contamination by other fungal or bacterial colonies, if contamination is present.

  • The titles of the figures should be rewritten in a way that only shows what the figure represents, rather than explaining the results and method as they currently exist. Examples: figures 1, 4, and 6. 
  • In the Conclusions, it would be better to expand further and add recommendations at the end.

Author Response

Comments 1: In the introduction: lines 101-104, does the genotype vary by region, or do the authors mean the presence of new genotypes? On the other hand, the authors mention that in China, the characterization of Fusarium fungi is poor, but China is a very large country (where in China? is better to specify), Or it is better to specify an environmental area rather than mentioning the entire country.

Response 1: Thank you for pointing out this issue. In lines 101–104 of the original introduction, our wording was indeed imprecise and may have led to misunderstanding.  Our original intention was to emphasize the discovery of new species from a taxonomic perspective, rather than to discuss regional genotypic variation. To avoid the ambiguity, we have revised the relevant sentence accordingly.

In addition, we fully agree with your suggestion that referring to “China” in general is too broad, given the country’s vast territory. Considering the actual sampling scope of this study, we have now specified the exact locations to avoid overly generalized statements.

These revisions have been made in lines 101–108 of the revised manuscript and are highlighted accordingly.

Comments 2: In lines 108-109: Here, the purpose of the study and the strategy used should be specified, rather than stating the results as they currently exist. Please amend.

Response 2: Thank you very much for your valuable comment. We fully agree that this section should clearly articulate the purpose of the study and the strategy employed, rather than prematurely presenting results. In response, we have revised the relevant portion of the introduction to better define the study's objectives and methodological framework. The updated content can be found in lines 101–108 of the revised manuscript and has been highlighted accordingly.

Comments 3: How was the purity of the fungal colonies determined after they were isolated from the three samples? The intent is to write the specifications of the pure fungal colonies to distinguish them from contamination by other fungal or bacterial colonies, if contamination is present.

Response 3: Thank you for your insightful comment. During the isolation process, the fungal strains were purified by a combination of repeated subculturing and microscopic examination to ensure colony uniformity and eliminate potential contamination. We have revised the relevant section in the manuscript accordingly; the updated description can be found in lines 125–129 of the revised version.

Comments 4: The titles of the figures should be rewritten in a way that only shows what the figure represents, rather than explaining the results and method as they currently exist. Examples: figures 1, 4, and 6.

Response 4: Thank you for your valuable comment. We fully agree with your suggestion that figure titles should concisely indicate what the figure represents, rather than describing results or methods. In response, we have revised the titles of Figures 1, 2, 3, 4, and 6 to align with this principle. For consistency, we referred to the formatting style used in the article “Morphological and Phylogenetic Analysis of a New Jellyfish of Phyllorhiza (Scyphozoa, Mastigiidae) from the East China Sea”, published in this journal. The revised titles are now included in the manuscript and have been highlighted accordingly.

Comments 5: In the Conclusions, it would be better to expand further and add recommendations at the end.

Response 5: Thank you for your comment. We fully acknowledge your suggestion. Accordingly, we have revised and expanded the Conclusions section. The revised content can be found in lines 815–836 of the revised manuscript.

Reviewer 2 Report

Comments and Suggestions for Authors

Dear Authors,

the manuscript reports a very huge work of charcterization of species of Nectriacecae. The issues in taxonomy and identification of this family are strongly relevant and the work done by the authors is appreciated.

As general comments, Enlish shoud be improved and the manuscript structuer appears confusing. Results and comments (discussion) shoul be clearly attributed to each section.

Some point by point comments follow:

Line 117: “Samples” should be in lowercase letter.

Line 117-119: rephrase the sentence

Line 125: “To” should be in lowercase letter. Monospore should be “monosporic”

Line 195: “for phylogenetic analyses” is redundant

Line 194-201: please, rephrase

Line 252: “examined” is repetitively used

Line 253: SNA and OA. Please, add manufacturer names

Results: share comments on results from results. Comments have to be included in the discussion

Line 399-448-501-547-595-654-702: it would be preferable to share results on microscopic observations from phylogenetic analyses. The comparison among results should be reported in the discussion

The Authors are strongly encouraged in improving the work description but the scientific content is suitable for publication.

Author Response

Comments 1: Line 117: “Samples” should be in lowercase letter.

Response 1: Thank you for your comment. We fully agree with your suggestion and have revised this instance accordingly. The change has been highlighted in line 116 of the revised manuscript.

Comments 2: Line 117-119: rephrase the sentence.

Response 2: Thank you for pointing this out. We fully agree with your suggestion and have rephrased the relevant sentences to improve their readability. The revised content appears in lines 116–122 of the revised manuscript and has been highlighted for your convenience.

Comments 3: Line 125: “To” should be in lowercase letter. Monospore should be “monosporic”

Response 3: Thank you for your comment. We fully agree with your suggestion and have revised these instances accordingly. These changes have been highlighted in lines 129 and 130 of the revised manuscript, respectively.

Comments 4: Line 195: “for phylogenetic analyses” is redundant

Response 4: Thank you for your comment. We fully agree with your suggestion and have removed the redundant expression from the manuscript accordingly.

Comments 5: Line 194-201: please, rephrase

Response 5: Thank you for your comment. We fully agree with your suggestion and have revised this section accordingly. The updated content has been highlighted in lines 206–213 of the revised manuscript.

Comments 6: Line 252: “examin ed” is repetitively used

Response 6: Thank you for your comment. We agree with your point and have replaced one occurrence of “examined” with “observed” to avoid redundancy. In addition, we have revised other repeated expressions in this section to improve readability. These changes are presented in lines 274–290 of the revised manuscript.

Comments 7: Line 253: SNA and OA. Please, add manufacturer names

Response 7: Thank you for your comment. We fully agree with your suggestion and have added the detailed formulations for SNA and OA media. The revised content has been highlighted in lines 275–278 of the revised manuscript.

Comments 8: Results: share comments on results from results. Comments have to be included in the discussion

Line 399-448-501-547-595-654-702: it would be preferable to share results on microscopic observations from phylogenetic analyses. The comparison among results should be reported in the discussion

Response 8: Thank you very much for your constructive comment. In taxonomic descriptions, it is a commonly accepted practice to include brief comments at the end of each species description, particularly to highlight distinguishing morphological and phylogenetic characteristics. This approach has been widely adopted in authoritative studies, such as:

  • Lombard, L. et al. (2015). Studies in Mycology, 80, 189–245.
  • Zhang, Y. et al. (2024). Mycosphere, 15, 6641–6717.
  • Crous, P.W. et al. (2021). Studies in Mycology, 98, 100116.
  • Zhang, H. et al. (2023). Mycosphere, 14, 2092–2207.
  • Han, S.L. et al. (2023). Studies in Mycology, 104(2), 87–148.

Considering the number of species described in this study, we have adopted this format to enhance clarity and avoid redundancy, while keeping the overall discussion section concise and readable. We have reviewed the relevant comments to ensure they effectively reflect both morphological traits and phylogenetic positions.

We hope this approach aligns with accepted practices and meets your expectations.

Comments 9: The Authors are strongly encouraged in improving the work description but the scientific content is suitable for publication.

Response 9: Thank you for your comment. In response to your suggestion, we have made overall improvements to the manuscript. The revised sections have been highlighted in the updated version for your convenience.

Reviewer 3 Report

Comments and Suggestions for Authors

The study by Fan et al. addresses species diversity and delimitation in the family Nectriaceae based on 22 isolates obtained from plant, fungal, and insect hosts in China, combining morphological characterization of conidiophores, macro- and microconidia, and sclerotia; multilocus phylogenetic analysis using conventional markers (ITS, TEF1-α, TUB2, RPB2); high-resolution phylogenomics of 4,941 single-copy orthologs; and divergence dating with molecular clocks. From to this integrated approach, the authors describe five new species (Fusarium dracaenophilum, F. puerense, F. wenshanense, Neocosmospora alboflava, and N. fungicola) and report two new host records for F. qiannanense and N. solani, demonstrating close congruence between multilocus and phylogenomic trees, which reinforces the robust taxonomic separation of Fusarium and Neocosmospora and provides a temporal framework with important implications for plant disease management, biodiversity conservation, and fungal evolution studies.

However, I have four methodological issues that should be clarified:

1) It is not explained why different clades were analyzed with different sets of gene regions rather than using a common panel across all isolates. Please clarify whether this decision was driven by unequal sequence availability, locus-specific evolutionary rates that optimize resolution in certain lineages, or phylogenetic representativeness criteria.

2) Readers less familiar with genomic methods may conflate a concatenation of half a dozen loci with a true phylogenomic analysis. I recommend adding a brief paragraph in Methods that defines a “multilocus” analysis versus a “phylogenomic” approach and explaining how these two strategies complement each other in this study.

3) In the divergence dating section, it should be made clear that the 20.54–86.06 Mya interval for the Geejayessia–Neocosmospora split is a secondary calibration from Lizcano Salas et al. (2024), which itself derives from the timescale of Lutzoni et al. (2018), according to my understanding. The authors should explicitly cite Lutzoni et al. (2018) as the primary source of their molecular-clock calibrations and briefly describe the 13 fungal fossils used to anchor key nodes in that phylogeny (for example, Palaeopyrenomycites devonicus, Devonian ≈ 400 Mya; Archaeomarasmius leggettii, Late Cretaceous ≈ 90 Mya; Gondwanagaricites magnificus, Early Triassic ≈ 250 Mya; etc.). They should then outline the datation workflow: Lutzoni et al. fixed those fossil ages in BEAST to generate a dated Ascomycota tree, and Lizcano Salas et al. applied a soft prior of 50–90 Mya in MCMCTree at the Fusarium–Neocosmospora node, yielding the 20.54–86.06 Mya credibility interval once combined with genomic rate data.

4) Although bootstrap values and posterior probabilities for the nodes defining the new species are high and imply their cohesion, I suggest to perform an explicit test of monophyly. The authors could generate two topologies—one unconstrained and one enforcing monophyly of each focal clade—calculate their log-likelihoods in IQ-TREE, for example, and then use CONSEL (or a similar program) to run Shimodaira–Hasegawa (SH) and Approximately Unbiased (AU) tests. This approach would provide quantitative evidence that no alternative topology fits the data significantly better.

Author Response

Comments 1: It is not explained why different clades were analyzed with different sets of gene regions rather than using a common panel across all isolates. Please clarify whether this decision was driven by unequal sequence availability, locus-specific evolutionary rates that optimize resolution in certain lineages, or phylogenetic representativeness criteria.

Response 1: Thank you for your insightful comment. The selection of different gene regions for different clades was primarily based on previous phylogenetic studies, such as Zhang et al. (2023) [1], Han et al. (2023) [2], and Wang et al. (2022) [3], where these lineages have been thoroughly investigated using lineage-specific loci. More fundamentally, the choice of gene regions was driven by two main factors: (1) variation in the evolutionary rates of loci across clades, which affects phylogenetic resolution, and (2) uneven sequence availability among different groups. This strategy aligns with previous research and is also clearly demonstrated in our Supplementary Figures S1–S5.

References (cited in response):

  1. Zhang, H., Zeng, Y., Wei, T., Jiang, Y., Zeng, X. (2023). Endophytic Fusarium and allied fungi from Rosa roxburghii in China. Mycosphere, 14, 2092–2207.
  2. Han, S.L., Wang, M.M., Ma, Z.Y., Raza, M., Zhao, P., Liang, J.M., Gao, M., Li, Y.J., Wang, J.W., Hu, D.M., et al. (2023). Fusarium diversity associated with diseased cereals in China, with an updated phylogenomic assessment of the genus. Studies in Mycology, 104, 87–148. https://doi.org/10.3114/sim.2022.104.02
  3. Wang, M., Crous, P.W., Sandoval-Denis, M., Han, S., Liu, F., Liang, J., Duan, W., Cai, L. (2022). Fusarium and allied genera from China: species diversity and distribution. Persoonia, 48, 1–53.

Comments 2: Readers less familiar with genomic methods may conflate a concatenation of half a dozen loci with a true phylogenomic analysis. I recommend adding a brief paragraph in Methods that defines a “multilocus” analysis versus a “phylogenomic” approach and explaining how these two strategies complement each other in this study.

Response 2: Thank you for your comment. We fully agree with your suggestion. Accordingly, we have added a brief paragraph defining multigene phylogenetic analysis and phylogenomic analysis in the revised manuscript. This addition can be found on lines 191–200 of the revised version and has been highlighted accordingly.

Comments 3: In the divergence dating section, it should be made clear that the 20.54–86.06 Mya interval for the Geejayessia–Neocosmospora split is a secondary calibration from Lizcano Salas et al. (2024), which itself derives from the timescale of Lutzoni et al. (2018), according to my understanding. The authors should explicitly cite Lutzoni et al. (2018) as the primary source of their molecular-clock calibrations and briefly describe the 13 fungal fossils used to anchor key nodes in that phylogeny (for example, Palaeopyrenomycites devonicus, Devonian ≈ 400 Mya; Archaeomarasmius leggettii, Late Cretaceous ≈ 90 Mya; Gondwanagaricites magnificus, Early Triassic ≈ 250 Mya; etc.). They should then outline the datation workflow: Lutzoni et al. fixed those fossil ages in BEAST to generate a dated Ascomycota tree, and Lizcano Salas et al. applied a soft prior of 50–90 Mya in MCMCTree at the Fusarium–Neocosmospora node, yielding the 20.54–86.06 Mya credibility interval once combined with genomic rate data.

Response 3: We sincerely appreciate the reviewer’s insightful suggestion. In the revised manuscript, we have clarified that the divergence interval of 20.54-86.06 Mya for the Geejayessia-Neocosmospora split represents a secondary calibration derived from Lizcano Salas et al. (2024), which is itself based on the dated fungal phylogeny constructed by Lutzoni et al. (2018). We now explicitly cite Lutzoni et al. (2018) as the primary source for the molecular-clock calibrations. In their study, 13 fossil constraints were fixed in BEAST to anchor key nodes in the Ascomycota phylogeny, including Palaeopyrenomycites devonicus (~400 Mya, Devonian), and Archaeomarasmius leggettii (~90 Mya, Late Cretaceous), genus Colletotrichum (~65.2 Mya, Upper Cretaceous). Based on this calibrated tree, Lizcano Salas et al. (2024) applied a soft prior of 50-90 Mya at the Fusarium-Neocosmospora node in their MCMCTree analysis, and the combination of this prior with genomic substitution rate data yielded the 20.54-86.06 Mya credibility interval used in our study. These clarifications have been added to the revised manuscript (lines 251–262), and the relevant references have been included in the bibliography.

Lizcano Salas, A.F.; Duitama, J.; Restrepo, S.; Celis Ramírez, A.M. Phylogenomic approaches reveal a robust time-scale phylogeny of the Terminal Fusarium Clade. IMA fungus 2024, 15, 13.

Lutzoni, F.; Nowak, M.D.; Alfaro, M.E.; Reeb, V.; Miadlikowska, J.; Krug, M.; Arnold, A.E.; Lewis, L.A.; Swofford, D.L.; Hibbett, D.; et al. Contemporaneous radiations of fungi and plants linked to symbiosis. Nature Communications 2018, 9, 5451, doi:10.1038/s41467-018-07849-9.

Comments 4: Although bootstrap values and posterior probabilities for the nodes defining the new species are high and imply their cohesion, I suggest to perform an explicit test of monophyly. The authors could generate two topologies—one unconstrained and one enforcing monophyly of each focal clade—calculate their log-likelihoods in IQ-TREE, for example, and then use CONSEL (or a similar program) to run Shimodaira–Hasegawa (SH) and Approximately Unbiased (AU) tests. This approach would provide quantitative evidence that no alternative topology fits the data significantly better.

Response 4: We sincerely appreciate your thoughtful suggestion. Based on previous taxonomic practices, the identification of new fusarioid fungi typically relies on fixed nucleotide differences from closely related taxa and stable morphological characteristics, supported by phylogenetic analyses such as maximum likelihood and Bayesian inference, along with their associated branch support values. This integrative taxonomic approach has been widely adopted and recognized in the literature, as demonstrated in several representative studies (e.g., Crous et al., 2021; Zhang et al., 2023; Han et al., 2023). In this study, we followed established taxonomic criteria by systematically comparing the nucleotide sequences and morphological traits between the proposed new species and known species, and constructed phylogenetic trees with topologies consistent with previous studies. The results show that the proposed species forms a highly supported monophyletic clade in the phylogenetic tree and exhibits clear morphological differentiation from its closest relatives. In addition, we incorporated Split tree analyses and phylogenomic data to further strengthen the evidence supporting the species' distinctiveness. Given the robust support from multiple lines of evidence, we believe that it is not necessary to conduct an additional constrained topology test for formal monophyly assessment at this stage. Nevertheless, we fully acknowledge the value of such analyses and will consider incorporating them in future work to further strengthen the rigor of our phylogenetic inference.

References (cited in response):

  1. Zhang, H., Zeng, Y., Wei, T., Jiang, Y., Zeng, X. (2023). Endophytic Fusarium and allied fungi from Rosa roxburghii in China. Mycosphere, 14, 2092–2207.
  2. Han, S.L., Wang, M.M., Ma, Z.Y., Raza, M., Zhao, P., Liang, J.M., Gao, M., Li, Y.J., Wang, J.W., Hu, D.M., et al. (2023). Fusarium diversity associated with diseased cereals in China, with an updated phylogenomic assessment of the genus. Studies in Mycology, 104, 87–148. https://doi.org/10.3114/sim.2022.104.02
  3. Wang, M., Crous, P.W., Sandoval-Denis, M., Han, S., Liu, F., Liang, J., Duan, W., Cai, L. (2022). Fusarium and allied genera from China: species diversity and distribution. Persoonia, 48, 1–53.
  4. Crous, P.W.; Lombard, L.; Sandoval-Denis, M.; Seifert, K.A.; Schroers, H.J.; Chaverri, P.; Gené, J.; Guarro, J.; Hirooka, Y.; Bensch, K.; et al. Fusarium: more than a node or a foot-shaped basal cell. Studies in Mycology 2021, 98, 100116